# Low modeled ozone production suggests underestimation of precursor emissions (especially NO$_x$) in Europe

Emmanouil Oikonomakis[1], Sebnem Aksoyoglu[1], Giancarlo Ciarelli[2], Urs Baltensperger[1] and André Stephan Henry Prévôt[1]

[1]Laboratory of Atmospheric Chemistry, Paul Scherrer Institute, Villigen, 5232, Switzerland
[2]Laboratoire Inter-Universitaire des Systèmes Atmosphériques (LISA), UMR CNRS 7583, Université Paris Est Créteil et Université Paris Diderot, Institut Pierre Simon Laplace, Créteil, France

*Correspondence to*: Sebnem Aksoyoglu (sebnem.aksoyoglu@psi.ch)

**Abstract.** High surface ozone concentrations, which usually occur when photochemical ozone production takes place, pose a great risk to human health and vegetation. Air quality models are often used by policy makers as tools for the development of ozone mitigation strategies. However, the modeled ozone production is often not or not enough evaluated in many ozone modelling studies. The focus of this work is to evaluate the modeled ozone production in Europe indirectly, with the use of the ozone–temperature correlation for the summer of 2010 and to analyze its sensitivity to precursor emissions and meteorology by using the regional air quality model, CAMx. The results show that the model significantly underestimates the observed high afternoon surface ozone mixing ratios ($\geq$ 60 ppb) by 10–20 ppb and overestimates the lower ones (< 40 ppb) by 5–15 ppb, resulting in a misleading good agreement with the observations for average ozone. The model also underestimates the ozone–temperature regression slope by about a factor of 2 for most of the measurement stations. To investigate the impact of emissions, four scenarios were tested: i) increased VOC emissions by a factor of 1.5 and 2 for the anthropogenic and biogenic VOC emissions, respectively, ii) increased NO$_x$ emissions by a factor of 2, iii) a combination of the first two scenarios, iv) increased only traffic NO$_x$ emissions by a factor of 4. For southern, central and eastern (except the Benelux area) Europe, doubling NO$_x$ emissions seems to be the most efficient scenario to reduce the underestimation of the observed high ozone mixing ratios without significant degradation of the model performance for the lower ozone mixing ratios. The model performance for ozone–temperature correlation is also better when NO$_x$ emissions are doubled. In the Benelux area, however, the third scenario (where both NO$_x$ and VOC emissions are increased) leads to a better model performance. Although increasing only the traffic NO$_x$ emissions by a factor of 4 gave very similar results as the doubling of all NO$_x$ emissions, the first scenario is more consistent with the uncertainties reported by other studies than the latter suggesting that high uncertainties in NO$_x$ emissions might originate mainly from the road-transport sector rather than other sectors. The impact of meteorology was examined with three sensitivity tests: i) increased surface temperature by 4˚C, ii) reduced wind speed by 50%, and iii) doubled wind speed. The first two scenarios lead to a consistent increase in all surface ozone mixing ratios, thus improving the model performance for the high ozone values but significantly degrading it for the low ozone values, while the third scenario had exactly the opposite effects. Overall, the modeled ozone is predicted to be

more sensitive to its precursor emissions (especially traffic NO$_x$) and therefore their uncertainties, which seem to be responsible for the model underestimation of the observed high ozone mixing ratios and ozone production.

## 1 Introduction

Surface ozone (O$_3$) has been identified as a threat to human health by causing respiratory problems (WHO, 2013; EEA, 2014) and can also cause damage to plants (Fowler et al., 2009). Tropospheric ozone is not directly emitted from a source, but it is a secondary pollutant formed by chemical reactions of other gases in the presence of sunlight in a complex, non-linear way (Monks, 2005). The main precursor species for ozone formation are the nitrogen oxides (NO$_x$ = NO + NO$_2$) and the volatile organic compounds (VOCs), which are emitted by various anthropogenic (e.g. industries, road vehicles, ships, etc.) and natural sources (e.g. plants, soil, etc.). Controlling these emissions therefore has been the main approach of ozone mitigation strategies (Monks et al., 2015). Apart from the ozone precursor emissions, the other key driver of the surface ozone concentrations, as well as its chemistry, is the meteorology; from local to global scale (Monks et al., 2015). For example, on the local scale changes in shortwave solar radiation and temperature can directly influence the ozone photochemistry, and changes in wind speed or vertical mixing can lead to accumulation or dilution of the ozone precursor concentrations as well as ozone itself. On the global scale, changes in atmospheric circulation patterns can influence the continental transport of ozone concentrations and its precursors, the stratosphere–troposphere ozone exchange and the local meteorology. As a large number of chemical and physical processes are involved in the formation and transport of tropospheric ozone, chemical-transport-models (CTMs) provide a useful tool for the investigation and assessment of the ozone concentrations as well as the processes influencing them.

The peak values of surface ozone concentrations usually occur in the summer afternoon hours when the temperature reaches its diurnal maximum and the incoming solar radiation is still ample. Since the very high ozone concentrations increase the risk for the damage to human health, as it happened for example, during the European heat wave in 2003 (Filleul et al., 2006), the understanding of ozone formation and reduction of risks is of primary interest. In order to better understand the role of drivers for ozone production and to introduce successful ozone mitigation strategies by means of CTMs, a consistent and careful model evaluation and data interpretation is required.

The evaluation of modeled ozone production by just comparing modeled ozone concentrations with measurements may be misleading, as an agreement between modeled and observed ozone concentrations might just be the result of compensating errors. On the other hand, it is known that surface ozone has a high positive correlation with temperature (Sillman and Samson, 1995; Pusede et al., 2015). As a result, temperature has been used in several studies (Neftel et al., 2002; Baertsch-Ritter et al., 2004; Bloomer et al., 2009) as a surrogate to indirectly assess surface ozone production via the ozone–temperature correlation. However, so far the use of the ozone–temperature correlation was only applied locally for individual stations and not at a greater regional scale. In this study we adopted alternative methods to assess the ozone concentrations, to unmask compensating errors and to evaluate the modeled ozone production in Europe. Furthermore, by applying

sensitivity tests we characterized the response of modeled ozone production to its two main drivers: emissions and meteorology.

The paper is organized as follows. In Sect. 2, the data and modelling methods are introduced, results are given in Sect. 3 beginning with model evaluation and then continuing with the evaluation of afternoon ozone mixing ratios, ozone production and its response to changes in model input such as emissions, meteorological parameters, initial and boundary conditions. Finally, conclusions are summarized in Sect. 4.

## 2 Methods

### 2.1 Model setup

In this study we used the regional air quality model, CAMx version 6.30 (comprehensive air quality model with extensions, http://www.camx.com). The modelling period covered the summer months (JJA) in 2010 with the last two weeks of May being used as spin-up time. The model domain extended from $15^{o}$W to $35^{o}$E and $35^{o}$N to $70^{o}$N in Europe with a horizontal resolution of 0.250º x 0.125º (Fig. 1). In order to perform a region-specific data analysis, the model domain was divided into 8 sub-regions, 7 of which are similar or identical to the PRUDENCE (http://ensemblesrt3.dmi.dk/quicklook/regions.html) climatic regions. The separation was also based on distinct local meteorological or chemical conditions such as in the Benelux area and the Po Valley in Northern Italy (Colette et al., 2012; Pernigotti et al., 2012, 2013; Thunis et al., 2015). We used 14 sigma layers going up to 460 hPa with the first layer being approximately 20 m thick. The concentrations are calculated at the mid-point of a given layer, so the modeled values of the first layer correspond to a height of approximately 10 m. Additional tests showed that higher vertical resolution with layers up to 100 hPa would have a negligible effect on surface ozone (see Fig. S1) as also shown by other studies (Menut et al., 2013; Markakis et al., 2015).

The gas phase mechanism used in this study was CB6r2 (Carbon Bond mechanism, version 6, revision 2: Hildebrandt Ruiz and Yarwood, 2013). We simulated the particle concentrations using CAMx's fine/coarse option. CAMx uses the ISORROPIA (Nenes et al., 1998, 1999) model for inorganic thermodynamics and gas–aerosol partitioning. We calculated the organic aerosol concentrations using the SOAP model (Strader et al., 1999). The calculation of dry deposition was based on the algorithms of Zhang et al. (2003). The initial and boundary conditions for the chemical species were obtained from the MOZART (Model of Ozone and Related Chemical Tracers) global model data for 2010 with a time resolution of 6 hours (Horowitz et al., 2003). These data were then interpolated to the size and resolution of our grid using the CAMx pre-processor MOZART2CAMx (RAMBOLL-ENVIRON, 2016). The photochemistry in CAMx is performed in two steps. First, clear-sky photolysis rates are calculated externally by the Tropospheric Ultraviolet and Visible (TUV) radiation model (NCAR, 2011) and then used as input into CAMx, where they are internally adjusted every hour for clouds, aerosols, pressure and temperature (Emery et al., 2010). In addition, for more accurate radiative transfer calculations, the 8-streams discrete ordinates scheme was used (Stamnes et al., 1988). Total Ozone Mapping Spectrometer (TOMS) data obtained by the

National Aeronautics and Space Administration (ftp://toms.gsfc.nasa.gov/pub/omi/data/) served as total ozone column input for both TUV and CAMx. The meteorological input and the emissions are discussed in detail in the next sections.

## 2.2 Meteorology

The meteorological parameters required as input for the air quality simulations were generated by the WRF-ARW model (Weather Research and Forecasting Model, version 3.7.1; Skamarock et al., 2008). The model domain and horizontal resolution were identical to those used for CAMx model (see section 2.1) while there were 31 vertical layers up to 100 hPa, of which 14 were selected for the CAMx runs for computational efficiency. The terrain and land use data were taken from 10´ data available from the United States Geological Survey (USGS). The selected key physical options for WRF parameterization are summarized in Table S1. Initial and boundary conditions for WRF were generated using 6 h European Centre for Medium-Range Weather Forecasts (ECMWF) re-analysis global data of resolution 0.72° x 0.72°. The same data were also used for four-dimensional data assimilation (FDDA) above the planetary boundary layer (PBL) in the WRF simulations. Moreover, the observational nudging in the meteorological simulations (i.e. the use of FDDA) has been shown to improve the prediction of ozone by the air quality models (Choi et al., 2009). The model was run as a 48 h forecast and was then re-initialized. The first 24 h were considered as spin-up and were discarded.

The WRF output was pre-processed with the WRFCAMx algorithm (RAMBOLL-ENVIRON, 2016) before being used by CAMx. The WRFCAMx pre-processor interpolates the meteorological variables from the WRF domain to the CAMx one (in our case only a vertical selection of the aforementioned 14 layers was done). Furthermore, it calculates vertical diffusivity ($K_v$) profiles (using the WRF planetary boundary layer height, PBLH), as the standard $K$-theory is applied in CAMx to account for vertical diffusion and sub-grid-scale mixing between layers. For the $K_v$ calculation the Yonsei University non-local closure scheme (YSU) PBL methodology was chosen to consistently match our WRF PBL parameterization. Finally, the minimum value for $K_v$ was set to 0.1 $m^2 s^{-1}$.

## 2.3 Emissions

We used the TNO-MACC-III European anthropogenic emission inventory for 2010 provided by the Netherlands Organization for Applied Scientific Research (TNO). The TNO-MACC-III is an extension of the TNO-MACC-II emission inventory (Kuenen et al., 2014) with some updates which are described in Kuik et al. (2016). It contains annual emission data for 10 SNAP (Selected Nomenclature for Air Pollution) categories per grid cell (Table S2). The TNO emission domain covers the same geographical space as our domain (Section 2.1) but with a higher horizontal resolution (0.125° x 0.0625°). By applying the monthly, weekly and diurnal profiles provided by TNO, we calculated the hourly gridded anthropogenic emissions of species required for CAMx. The total $NO_x$ and NMVOC (non-methane volatile organic compounds) emissions per SNAP category in summer 2010 are shown in Fig. 2. The inventory does not include sea salt, mineral dust, wild fire emissions and NO emissions from lightning. The air quality simulations, however, do contain sea salt and mineral dust aerosol concentrations from the initial and boundary conditions.

The biogenic emissions (isoprene, monoterpenes, sesquiterpenes, soil NO) were calculated according to the method described by Andreani-Aksoyoglu and Keller (1995) using temperature, shortwave solar radiation and USGS land use data from the WRF output and the GlobCover 2005-06 inventory (http://due.esrin.esa.int/page_globcover.php). Spatial distribution maps of those biogenic emissions are provided in the Supplement (Fig. S2). All emissions were treated as area

emissions in the first model layer. Uncertainties in the emission estimates vary depending on the emitted pollutants and their sources (Kuenen et al., 2014). Among the anthropogenic emissions, one of the most important contributors, with high uncertainty, is the road transport (SNAP 7) which was shown to be the category with the highest contribution to the daily average maximum 8 h ozone mixing ratio in Europe (Tagaris et al., 2015). The uncertainty in $NO_x$ and NMVOC emissions from road transport was rated as C (C corresponds to a typical error range of 50 to 200%) by the European Environment

Agency (EEA, 2016). Especially high uncertainty in $NO_x$ emissions from the diesel vehicles might be related to non-compliance with air quality regulations or insufficiencies in the air quality regulation control. For example, in several studies emissions from passenger cars were measured in different, more realistic driving conditions than the laboratory test New European Driving Cycle (NEDC) (Hausberger, 2010; Weiss et al., 2011a, b, 2012; Alves et al., 2013; May et al., 2013). These studies showed that there was a significant discrepancy (a factor of 2–4) in the $NO_x$ emissions from light-duty diesel

vehicles between the two driving cycles, indicating inadequacy of the NEDC to effectively control the compliance of passenger cars with the European air quality regulations. As a consequence, large discrepancies have been observed between real-world emissions of diesel passenger vehicles based on remote sensing and simultaneous license plate detection at a road site in Switzerland, and the homologation limit of diesel passenger vehicle (Baltensperger, 2016). According to Anenberg et al. (2017), also the heavy duty diesel trucks and buses emit more $NO_x$ than the legislative limit. Furthermore, Vaughan et al.

(2016) and Karl et al. (2017) reached similar conclusions by analyzing $NO_x$ flux measurements and attributed the discrepancy between observations and emission inventory estimates to the under-representation of the real-world road traffic emissions. Moreover, a general underestimation of the total $NO_x$ emissions, compared to TNO MEGAPOLI and MACC-III inventories, by a factor of 1.4-1.5 for the summer of 2009 in Paris was recently reported by Shaiganfar et al. (2017), where they used a large set of car multi-axis differential optical absorption spectroscopy (MAX-DOAS) measurements to calculate

the $NO_x$ emissions by applying the closed integral method (CIM). For the VOC emissions there are reported emission uncertainties of ~50% for the anthropogenic sources (Theloke and Friedrich, 2007; Kuenen et al., 2014). The VOC emission uncertainties can be due to a number of reasons such as: i) the small number of measured vehicles for the transportation sector, since the VOC species resolution rely on measurements, ii) not enough available measurement data for the combustion-, process-, and production-related emissions compared to the much higher number of individual emission

sources, iii) the large variety of the VOC compositions in the used solvents, iv) the measurement uncertainties (Theloke and Friedrich, 2007). Biogenic VOC emission estimates on the other hand, have higher uncertainties (a factor of 2–3) associated with their transformation in the atmosphere and the lack of sufficient measurements of biogenic species (Karl et al., 2009; Hogrefe et al., 2011; Oderbolz et al., 2013; Guenther, 2013). In addition, the marine transport sector is one of the least regulated anthropogenic emission sources with emissions from ships having high uncertainties (EEA, 2016) and can have an

important contribution to surface ozone in the Mediterranean sea, coastal areas and to some extent over land (Tagaris et al., 2015, 2017; Aksoyoglu et al., 2016).

## 2.4 Observations

Meteorological observations from European stations with 3 h time intervals were obtained from the British Atmospheric
Data Centre (BADC) using the UK Met Office Integrated Data Archive System (MIDAS) Land Surface Stations database (Meteorological Office, 2013). Even though the UK stations have hourly observations, for the sake of a more homogeneous and consistent model performance evaluation for the whole European domain the 3 h interval was used for the UK stations as well. The extracted meteorological parameters were: dewpoint and air temperature at 2 m (T), wind speed and direction at 10 m (WS and WD, respectively) and surface air pressure. The water vapor mixing ratio ($q_v$) was calculated using the
dewpoint temperature and surface air pressure as described in the literature (Bolton, 1980; Wagner and Pruß, 2002). Only stations that belong to the synoptic network (SYNOP) were used for the WRF performance evaluation, as only those stations meet the requirements for forecasting as given in the MIDAS user's guide (http://badc.nerc.ac.uk/data/ ukmo-midas/ukmo_guide.html) and therefore contain data appropriate for comparison with the instant WRF output values. All data are reported in UTC time.

There are no direct measurements of the PBLH, but it can be estimated with different methods by using sounding data. Such data were extracted from the University of Wyoming database (http://weather.uwyo.edu/upperair/sounding.html). All 79 sites have one sounding at 12:00 UTC and most of them have also a second one at 00:00 UTC. Since not all sites have soundings at 00:00 UTC and the concept of the PBLH applies only for convective periods, only the soundings at 12:00 UTC were selected for evaluation. We used the bulk Richardson number ($Ri_{bc}$) method to estimate the PBLH above ground, which
is considered as the altitude where the $Ri_{bc}$ exceeds a critical value $Ri_{cr}$ (Seibert et al., 2000). Although there is a range of values for $Ri_{cr}$ proposed in the literature (Richardson et al., 2013; Zhang et al., 2014) we selected the $Ri_{cr}$ to be 0.25 for both stable and unstable conditions which is also used in the PBLH calculations with the YSU scheme in WRF (Hong, 2010). The same method and the critical value were also used in other air quality modelling studies for PBLH evaluation (Brunner et al., 2015; Bessagnet et al., 2016).

The observational data for the surface air pollutant concentrations (http://acm.eionet.europa.eu/databases/) were taken from the European Air Quality Database v7 (AirBase; Mol and De Leeuw 2005). In order to reduce the uncertainty due to grid resolution we used only background rural stations with hourly (UTC) measurements for comparison with the model output. The chemical species used in the evaluation are: $O_3$, $NO_2$, $SO_2$, CO and $PM_{2.5}$. In addition, we used ozonesonde data from the World Ozone and Ultraviolet Radiation Data Centre (Toronto, Canada; http://woudc.org/data.php) for 6 sites to evaluate
the vertical profiles of ozone, temperature and wind speed (discussed in Sect. 3.3). A short description of the ozonesonde stations is given in Table S3. Finally, data quality filters were applied to exclude surface stations with less than 90% data availability and with elevation higher than 700 m. For the radiosonde sites a less strict filter of 2/3 data availability was applied due to the low measurement frequency.

## 2.5 Model evaluation methods

For comparison with surface observations the values in the lowest model layer were interpolated (bilinear interpolation) to each station's coordinates, while for the evaluation of vertical profiles the nearest neighbor method was used for horizontal interpolation together with linear vertical interpolation to 14 constant heights above ground. The statistical metrics that were
used for the meteorological and air quality model performance evaluation are summarized in Table 1. The statistical metrics for the wind direction were calculated only for wind speeds higher than 1.5 m s$^{-1}$ to omit the high observational errors below this threshold (Zhang et al., 2013). For the meteorological parameters the model evaluation was performed for the respective available time interval, while for the chemical species the evaluation was done for the daily mean values in order to be comparable with other studies using other models and parameterizations (e.g. Bessagnet et al., 2016). We calculated the
daily means from the hourly measurements to ensure that it corresponds to the time range of 00:00–23:59 for the day. As this study focuses on ozone, additional evaluation of its diurnal variation and afternoon (when most of ozone production takes place) mean was performed, as well as for NO$_2$ since it is one of the main precursors for ozone formation. The analysis of each statistical metric was first performed for each station individually (to avoid spatial noise) and then the total mean of all stations was taken as the representative value of the model performance evaluation for the whole domain. The statistical
results were also compared with recommended model performance criteria for model evaluation, which are shown in Table 2.

In addition to the aforementioned traditional evaluation methods, we used other, less common, approaches for the evaluation of modeled ozone in our study. We applied these non-traditional methods in the afternoon hours (12:00–18:00 UTC; only 12:00, 15:00 and 18:00 UTC for the meteorology) when the ozone production and mixing ratios often reach their maximum.
For the evaluation of ozone mixing ratios, we divided the observed values into mixing ratio bins of 10 parts per billion by volume (thereafter ppb) between 20 and 70 ppb, plus one bin incorporating all the values equal or higher than 70 ppb. For each observed ozone mixing ratio bin we calculated the mean bias (as defined in Table 1) between the respective model values and observations. This approach shows and quantifies more clearly the model's prediction for each respective observed value set, avoiding compensation of errors on the temporal scale. This greatly improves the interpretation of the
model's prediction, especially if it is to be compared with other models or sensitivity tests.

The evaluation of ozone production was performed indirectly, with the use of its correlation with temperature as discussed in Sect.1. We made use of the ozone–temperature correlations as described in the following three approaches:

1) We selected 8 surface stations (see Table S4 for details), which have measurements of both temperature and ozone, and performed regression analysis (use of scatter plot) between afternoon mean ozone mixing ratios and the respective afternoon
mean temperature for each station. Since we used different measurement networks for the air quality and meteorology, the characterization of a station as common in both networks was based on the very small difference (< 0.01°) of the station's reported coordinates (both longitude and latitude) between the two networks. The next step was to identify a linear relationship between the ozone and temperature values and apply a best linear fit. Since the least-square linear regression

method can be sensitive to outliers, we used a more robust linear regression technique: the Theil-Sen estimator (Sen, 1968). From the best linear fit we calculated the slope that represents the ozone production as a function of temperature. By comparing these slopes with the ones from the modeled ozone and temperature, we evaluated the modeled ozone production.

2) In order to evaluate the model results using all stations with ozone data (in the first step, we could use only 8 stations which had both ozone and temperature measurements) we applied an additional method. We compared the observed ozone–temperature correlation with the correlation between observed ozone and modeled temperature. This was done to assess and confirm (together with the meteorological model evaluation in Section 3.1) that the modeled temperature was a good surrogate for the observed temperature in the ozone–temperature correlation. In this way, we could apply this method to all stations and evaluate the ozone production in the whole European domain. It is difficult, however, to interpret the results when the evaluation is performed for each station separately when the number of stations is large. We displayed therefore all the calculated slopes of the ozone–temperature linear fit for both observations and model into a single scatter plot. In this way, the illustration and interpretation of the modeled ozone production evaluation for whole domain became simpler. In addition, for more consistent results two filters were applied in the method above: i) we only included days with afternoon mean temperature higher or equal to 15˚C, ii) since stations in colder regions do not have very high temperatures even in summer, we only kept stations with at least 2/3 data availability (after the first filter was applied).

3) In order to have a more rigorous model evaluation of the ozone production without the influence of day-to-day variation and local meteorological conditions, we also applied a binned data analysis in the ozone–temperature correlation as also used by Bloomer et al. (2009). We divided the modeled temperature into four bins with 5˚C intervals starting at 15˚C and ending at temperatures equal or higher than 30˚C. For each temperature bin the mean ozone mixing ratio for the respective values was calculated. With this third approach a more general picture (representative for each region) of the ozone–temperature regression is shown. All three approaches comprise the core of the modeled ozone production evaluation of this study and will also help apportion its potential errors, as correctly as possible, to its sources. A prerequisite of these methods' consistency is a good meteorological model performance which is evaluated in Sect. 3.1 along with the air quality model results.

**2.6 Sensitivity tests**

In order to characterize the sensitivity of the modeled ozone production to its main drivers, various emission and meteorological sensitivity tests were performed (see Table 3). These tests were based on the emission uncertainties that were discussed in Sect. 2.3 as well as the meteorological uncertainties of this study such as temperature and wind speed underestimation and overestimation of low-wind speed, which are quite common in modelling studies (Solazzo et al., 2013, 2017; Im et al., 2015; Bessagnet et al., 2016).

**3 Results and discussion**

**3.1 Model performance evaluation**

The meteorological model results show a good agreement with the surface observations for 1051 stations (Table 4) and meet the performance criteria (Table 2) suggested by Emery et al. (2001). Only the mean gross error (MGE, see Table 1 for definitions) for the wind direction is slightly off by 5 deg. Apart from the surface meteorological parameters, also the PBLH (56 stations) is predicted quite well with a high index of agreement (IOA), and the mean bias (MB) and root-mean-square error (RMSE) are well within the range of other studies (Brunner et al., 2015; Bessagnet et al., 2016).

The overall model performance for the daily mean concentrations of the air pollutants in summer (JJA) 2010 (Table 5) was reasonably good. The statistical evaluation results for most chemical species were in line with those reported for various models and parameterizations for summer periods in Europe (Bessagnet et al., 2004, 2016; Solazzo et al., 2012b, b; Nopmongcol et al., 2012; Giordano et al., 2015). Model performance goals and criteria for $O_3$ and $PM_{2.5}$ (Table 2), recommended by Boylan and Russell (2006) and EPA (2007), were met. Moreover, $O_3$, which is the main focus of this study, was only slightly over-predicted by 4 ppb and had a high correlation coefficient (*r*) of 0.7. On the other hand, $SO_2$ is overestimated with a MB and RMSE of 1 and 2 ppb, respectively. In the EURODELTA III model inter-comparison exercise, models showed the worst performance for $SO_2$ (Bessagnet et al., 2016). Possible reasons for this behavior, as also discussed in Ciarelli et al. (2016), can be the injection height of the $SO_2$ emissions from high stack point sources which are placed in the first model layer (i.e. up to 20 m), especially near the harbors and coastal areas, as well as insufficient conversion to sulfate and deposition processes. The CO concentrations were underestimated (MB and MGE were close in absolute terms and correlation was poor). However, the accurate modelling of CO is a common problem in the European modelling community and our results are similar to other studies (Nopmongcol et al., 2012; Solazzo et al., 2013, 2017; Giordano et al., 2015). Since CO concentrations do not change rapidly by chemistry and deposition processes, the differences between model and observations are mostly related to boundary conditions, vertical mixing and emissions (Solazzo et al., 2013, 2017; Giordano et al., 2015). Although the bias for $NO_2$ is small (–0.2 ppb), the MGE and RMSE are much higher (in absolute terms) indicating compensation between over- and underestimation throughout the day leading to a weak correlation coefficient (0.4). The largest discrepancies occur in the night and early morning hours ($NO_2$ diurnal profile is discussed in detail below). The model performance for $PM_{2.5}$ looks good (small negative MB), however, a similar compensation of errors as in the case of $NO_2$ appears to occur for $PM_{2.5}$ concentrations as well. Since $NO_2$ and $SO_2$ are precursors for the $PM_{2.5}$ formation, their errors (especially in the night and early morning hours) are expected to affect the $PM_{2.5}$ concentrations in a similar way. In addition, the lack of wildfire emissions could also contribute to the discrepancies between model and observations for $PM_{2.5}$ and CO (Saarikoski et al., 2007; Hodzic et al., 2007; Tressol et al., 2008; Turquety et al., 2009; Strada et al., 2012).

The diurnal profiles of $O_3$ and $NO_2$ for each region are shown in Fig. 3 and 4, respectively. The model captures quite well the $O_3$ diurnal variation, especially in the afternoon for most regions except for the Po Valley (PV region) where models have

usually difficulties in this heavily polluted area with complex topography (de Meij et al., 2009a), and the British Isles (BI region) where there is a consistent slight overestimation. The overestimation during the night and early morning hours can be due to overestimation of vertical mixing, which causes stronger vertical transport of $O_3$ from the higher altitudes to the surface and thus enhancing the surface mixing ratios (Lin et al., 2008; Lin and McElroy, 2010; ENVIRON, 2011). The effect

is the opposite for $NO_2$, where more mixing during the night and early morning hours results in enhanced transport of $NO_2$ from the surface to the upper layers leading to lower $NO_2$ mixing ratios in the lower layers. However, there can be different level of uncertainty in the $K_v$ values for different layers and thus different effects on $NO_2$ mixing ratios, especially in the first layer where the emissions are injected (ENVIRON, 2011). In addition, the nocturnal dilution of $NO_2$ will also impact the night time $NO_x$ titration of ozone and this will influence both the mixing ratios of $NO_2$ and $O_3$. The early morning peak in the

$NO_2$ diurnal profile is related to the traffic $NO_x$ emission peak where there is a time shift of one hour between the model and observations. This is probably due to very low $K_v$ values in those early morning hours for the first model layer, which confine the emissions to the surface (ENVIRON, 2011). Since the $NO_x$ emissions are not efficiently transported out of the first model layer, they lead to a peak of $NO_2$ mixing ratios one hour earlier than the $NO_x$ emissions' early morning peak ($NO_x$ emissions are already high one hour before their peak time). The same source of error could also account for the

overestimation of evening surface $NO_2$ mixing ratios in most regions. Other sources of error for the $NO_2$ mixing ratio during the night-early morning hours can be related to uncertainties in its dry deposition (Simpson et al., 2014) or to the coarse grid resolution (some background rural stations might be located in grid cells that are characterized by urban conditions). On the other hand, the model underestimates the $NO_2$ in the afternoon by up to a factor of ~ 2 for all regions apart from the Po Valley (PV region), where it is even higher. It is known that the observed $NO_2$ mixing ratios, which are mainly measured

with instruments equipped with molybdenum converters, can be overestimated due to instrumental artifact. Steinbacher et al. (2007) reported that in the summer afternoon hours for a non-elevated rural site in Switzerland the ratio of $NO_2$ mixing ratios measured with molybdenum converter to the ones measured with photolytic converter (i.e. without that artifact) was on average ~1.7. However, this overestimation in the $NO_2$ observations cannot solely explain the model's afternoon under-prediction as it is higher than the measured $NO_2$ artifact, as indicated by the diurnal variation of the ratio of observed to

modeled $NO_2$ mixing ratio for the base case (Fig. S3). The rest of this discrepancy can be mainly attributed to emission and/or meteorological uncertainties.

In order to investigate the afternoon $O_3$ and $NO_2$ mixing ratios in more detail, we analyzed the afternoon averaged (12:00–18:00 UTC) scatter plots (Fig. 5 and 6). The good agreement between modeled and measured afternoon ozone in Fig. 3 seems to be the result of a compensation of errors. More specifically, in the afternoon the model mainly over-predicts the

low ozone mixing ratios ($\leq 40$ ppb) and under-predicts the high ones ($\geq 50$ ppb), especially in central Europe (PV, ME and BX regions). While the overestimation of the lower observed ozone values is more likely linked to transport (vertical and horizontal) processes, the underestimation of the higher ones might be an indication of underestimation in ozone production. Similar model bias patterns as in this study were also reported by other studies for a variety of different models and parameterizations in Europe, the vast majority of which showed overestimation of the low ozone concentrations and

significant underestimation of the high ozone levels (Solazzo et al., 2012b; Im et al., 2015). In the less polluted SC and BI regions most of the observed ozone values do not grow above 60 ppb (98-99% of the sample) and so the region is mainly characterized by the overestimation of the lower ozone values. On the other hand, the afternoon bias in the $NO_2$ mixing ratios (underestimation by factor of 2 for the whole domain except for the Scandinavia (SC region)) is consistent with the diurnal plots (Fig. 4) and appears to be more pronounced (Fig. 6). However, for BI and SC regions the $NO_2$ results should not be interpreted as a robust representation of the whole region due to the small number of sites (4 and 3 respectively) that are included.

As the afternoon ozone mixing ratios are strongly related to ozone production, we made use of the ozone–temperature correlation (as discussed in Section 2.5) to examine the modeled ozone production performance. The regression between surface afternoon mean ozone mixing ratio and temperature for 8 stations is shown in Fig. 7. Three cases are shown: i) observed ozone mixing ratios against observed temperature, ii) observed ozone mixing ratios against modeled temperature, and iii) modeled ozone mixing ratios against modeled temperature. For all cases a strong linear correlation of ozone with temperature with an upward trend is evident, except for the Nice (FR) station where ozone stays constant with increasing temperature. A comparison of the ozone–temperature correlation for the first two cases (black and red colors) shows that the modeled temperature can be used consistently as a surrogate for the observed one and can therefore be paired with the observed ozone mixing ratios. For the third case (blue color), the upward trend of the ozone–temperature correlation is less steep compared to the other two cases. This is mainly due to the underestimation of the high ozone mixing ratio values ($\geq 60$ ppb). Since the ozone–temperature correlation is a proxy for the ozone production performance we can argue that the model underestimates the ozone production at these stations.

In general, the use of daily means and diurnal profiles for the model performance evaluation may conceal hidden biases as shown above. Especially for a chemical species like ozone, which is greatly influenced by both the meteorology and its complex non-linear chemistry, a model evaluation should be carried out for hourly values to increase the evaluation's consistency but also to better examine and understand the physical and chemical processes leading to the modeled values. Regarding the ozone production, the use of the afternoon ozone–temperature correlation indicated an underestimation of the model, but it was limited to 8 stations only. In the next sections we employ the rest of the methods discussed in Section 2.5 on all stations to better evaluate both qualitatively and quantitatively the model afternoon ozone mixing ratio and production, and apply various sensitivity tests to investigate the sources of error.

### 3.2 Sensitivity of ozone to emissions

**Base case:** Figure 8 shows the mean bias in the modeled afternoon ozone mixing ratios as a function of measured ozone mixing ratio bins (as discussed in Section 2.5) for the base case as well as for four emission scenarios described in Table 3. The trend of the model bias for the base case is very similar to the one in Fig. 5: in all regions afternoon ozone mixing ratios higher than or equal to 50 ppb are underestimated (3–17 ppb) and this underestimation increases with the mixing ratio. In the Po Valley (PV region), which has the largest number of measurement data in the highest mixing ratio bin ($\geq 70$ ppb), the

mean negative bias is about 15 ppb. The lower afternoon ozone mixing ratios (< 50 ppb) are overestimated in the whole domain with more regional variations than in the case of higher mixing ratios (≥ 50 ppb). The only exception to this overestimation appears in the less polluted regions BI and SC, where the overestimation in the lower bins (< 50 ppb) is either very small or close to zero. More specifically, the positive model bias is higher for the stations in southeast France as well as

south and central Italy (MD region) with up to 20 ppb for the first bin (20–30 ppb) and then gradually decreasing with increasing mixing ratio. With increasing latitude the positive bias is reduced and reaches almost zero at the stations in BI and SC regions. In general, the low afternoon ozone mixing ratios (< 50 ppb) at the background rural sites are more likely related to background ozone levels and influenced more by the meteorology. On the contrary, the higher afternoon ozone mixing ratios (≥ 50 ppb) are usually associated with ozone production, where the ozone precursors and thus the emissions play a key

role. This is confirmed by the various emission sensitivity tests we applied. However, the meteorology can influence the mixing ratios of ozone precursors by vertical mixing or advection, especially for sites that are located downwind of high emission areas.

**Increased VOC emissions**: The model's response to increased VOC emissions (1.5-2VOC scenario, Table 3) is relatively weak for most of the regions except for MD, PV and BX regions (Fig. 8) with the largest effect of ~4 ppb reduction of the

negative bias occurring in the highest bin (≥ 70 ppb). Moreover, for the lowest three bins the effect is negligible in regions IP, MD, EA and BI. A higher impact is seen in the polluted areas such as the Po Valley (PV region), the Mediterranean coasts in Italy and southeast France (MD region) and the Benelux area (BX region). The Benelux area is exposed to high $NO_x$ emissions from both land and shipping activities, leading to a more VOC sensitive chemical regime for ozone production in this region (Beekmann and Vautard, 2010; Aksoyoglu et al., 2012). The geographical characteristics of the Po

Valley in Northern Italy lead to a trap and accumulation of the pollutants in the area (de Meij et al., 2009a, b; Pernigotti et al., 2012, 2013), which in return can also affect the nearby stations that are located in the MD region. For both PV and BX regions there is a consistent increase in modeled ozone mixing ratios for all bins resulting in a decrease in the negative bias in higher bins and a slight increase in the positive bias in lower bins (Fig. 8).

**Increased $NO_x$ emissions**: A larger impact on the ozone mixing ratios (negative bias improved by ~6-8 ppb) is observed

with increased $NO_x$ emissions ($2NO_x$) for all regions except for BX and BI regions. In BX region the higher bins were not affected while mixing ratios in the lower bins decreased most likely due to more titration, consistent with the VOC sensitive regimes. The ozone mixing ratios in the BI region were insensitive to the increase of the $NO_x$ emissions, as background levels mainly govern ozone levels in that area. On the other hand, there was a small enhancement (up to ~2.5 ppb) of the positive bias for the lower ozone mixing ratios (< 50 ppb) in IP, MD, EA and SC regions. The effect of increasing only the

traffic $NO_x$ emissions by a factor of 4 (4traf_$NO_x$ scenario) is very similar to the $2NO_x$ scenario. It reduces the negative bias slightly more (~1–2 ppb) compared to the $2NO_x$ scenario in the two highest bins (≥ 60 ppb) in regions EA and ME without increasing the overestimation in the lower bins (<50 ppb). Only in the Po Valley (PV region) is the model's response slightly weaker (~2–3 ppb) for the two highest bins (≥ 60 ppb) compared to the $2NO_x$ scenario, where negative bias was not reduced as much as with the $2NO_x$ scenario. However, this might be related to enhanced ozone titration by $NO_x$, as the positive bias

in the lower bins ($< 50$ ppb) decreased more ($\sim$3–4 ppb) than with the $2NO_x$ scenario. Finally, since the $4traf\_NO_x$ scenario has a very similar impact on surface ozone as the $2NO_x$ scenario and it is within the reported observed underestimation range (i.e., factor of 2–4; see Sect. 2.3), this might suggest that high uncertainties in the $NO_x$ emissions might be more relevant to the road-transport sector (SNAP 7; see Fig. 2).

**Increased $NO_x$ and VOC emissions:** The combined increase of both $NO_x$ and VOC emissions has the largest impact among all the emission scenarios. For all regions (except for BX region), the ozone mixing ratios consistently increase in all bins leading to an underestimation only for the highest ozone levels and overestimation for all other bins. For BX region, this scenario reduces the bias in all bins. In the lower ozone mixing ratio bins ($< 40$ ppb) the $NO_x$ emissions are responsible for the ozone destruction causing the reduction of the positive bias, while in the higher ozone mixing ratio bins ($\geq 50$ ppb) the
enhancement of ozone production leads to a reduction of the negative bias by 2–7 ppb (negative bias reduction increases with ozone mixing ratio). For the 40–50 ppb bin, there is a negligible change ($< 1$ ppb) towards a negative bias.

In general, the PV region exhibits the highest sensitivity to emissions due to its location, and the model prediction for ozone is generally improved with the increased $NO_x$ emissions ($2NO_x$ and $4traf\_NO_x$ scenarios). For the rest of the Southern European stations (IP and MD regions), the increase of the $NO_x$ emissions ($2NO_x$ and $4traf\_NO_x$ scenarios) also gives the
best results but the overall modeled ozone performance remains problematic as the overestimation of the lower ozone mixing ratios ($< 50$ ppb) is enhanced (in a smaller degree for the $4traf\_NO_x$ scenario in MD region) without effectively tackling the underestimation problem of the higher ozone mixing ratios ($\geq 60$ ppb). Similarly for central Europe (ME region), increasing the $NO_x$ emissions ($2NO_x$ scenario) and especially the transportation $NO_x$ emissions ($4traf\_NO_x$ scenario) improves the base case more than any other emission test by reducing the negative bias for high ozone mixing ratios ($\geq 50$ ppb) and having
only a small bias (positive or negative) for other ozone mixing ratio ranges. On the other hand, increasing both $NO_x$ and VOC emissions ($2NO_x$,1.5-2VOC scenario) has the most effective improvement in the model performance for the BX region, since it is the only case where the ozone bias decreases in all bins. The British Isles (BI) and Scandinavia (SC) are the only regions where the base case performs quite well, with only a $\pm 5$ ppb or less bias for ozone mixing ratios less than 60 ppb which comprise 98-99% of the total ozone mixing ratio range for those regions. Although increased VOC emissions
improve the results slightly, the change is very small. Overall, our emission-sensitivity analysis indicates that the $NO_x$ emissions, especially from the transportation sector (SNAP 7) in central, eastern and southern Europe might be too low in the emission inventories.

**Ozone–temperature correlation:** We analyzed the slopes of the regression lines from the ozone–temperature correlations using the second approach, as described in Section 2.5. The modeled slopes are displayed as a function of observed slopes
for each region and for each emission scenario in Fig. 9. For the base case, the model underestimates the ozone–temperature slope by about a factor of 2 or more for most stations in all regions apart from the BI and SC regions (light blue and purple color, respectively), where the most stations are close to the 1:1 line. The underestimation of the slopes is more evident for the IP and MD regions (yellow and pink color, respectively). For the MD region, despite the underestimation of the high ozone mixing ratios, the model also overestimates the low ozone mixing ratios more significantly than for other regions (see

Fig. 8) and this will consequently influence the trend of the ozone–temperature regression. Increasing the VOC emissions (1.5-2VOC scenario) does not change the picture compared to the base case with the exception of improvement in the BX region (red color), which is consistent with the results shown in Fig. 8. On the other hand, the scenarios with increased $NO_x$ emissions ($2NO_x$ and $4traf\_NO_x$ scenarios) as well as with increased $NO_x$ and VOC emissions ($2NO_x$,1.5-2VOC scenario)

improve the modeled ozone–temperature slopes. The difference between these two cases (increasing only $NO_x$ or both $NO_x$ and VOC emissions) is mainly for the BX region, where the $2NO_x$,1.5-2VOC scenario performs better (red dots get closer to the 1:1 line), which is again consistent with the aforementioned analysis of Fig. 8. The same scenario may also bring some stations of other regions closer to the 1:1 line, but by combining the results from Fig. 8 one can see that it overestimates the ozone values in all bins except for the last one. This underlines the need of an additional approach to evaluate the ozone–

temperature correlation by taking into account both the regression slope and the magnitude of ozone mixing ratios.

The ozone–temperature correlation was also investigated from a different perspective smoothing out the station-to-station variation by making use of the third approach discussed in Section 2.5. The results are shown in Fig. 10 where the 15˚C temperature threshold cuts off most of the lowest afternoon ozone mixing ratios (< 30 ppb) as the mean ozone mixing ratios (modeled and observed) in the 15–20˚C bin are greater or equal to 35 ppb for all regions and the production of ozone is

likely very low for temperatures lower than this threshold. This allows accentuating on the underestimation of the high ozone mixing ratios which is more relevant for the ozone production. Furthermore, the advantage of Fig. 10 is that it summarizes information from both Fig. 8 and 9: the height of the bars depicts the underestimation/overestimation of the high/low ozone mixing ratios, while the trend of their relationship with temperature (which is more clearly illustrated by the lines above them) represents an evaluation of the model performance for ozone production. By looking at both of these characteristics in

Fig. 10, the modeled ozone production for the base case is under-predicted in all regions apart from BI and SC where it is in good agreement or slightly over-predicted. More specifically, the observed ascending trend (black line) is stronger for the MD, PV, ME and BX regions compared to the base case, while for the IP and EA regions the observed trend is weaker and closer to the base case and especially in the IP region it starts to level off at high temperatures (≥ 30˚C). The difference between the base case (red color) and the 1.5-2VOC scenario (green color) is the smallest among all the emission sensitivity

tests in all temperature bins. The $2NO_x$ (blue color), $4traf\_NO_x$ (yellow color) and $2NO_x$,1.5-2VOC (purple color) scenarios have very similar trends (with almost parallel lines) for all regions, but they differ in the height of the bars (i.e. the ozone mixing ratio values) with the exception of the BI and SC regions where they are also similar. By considering both bar height and line trend, increasing just the $NO_x$ emissions ($2NO_x$ or $4traf\_NO_x$ scenario) improves the model performance for ozone production for the MD, PV and ME regions, while for the IP and EA regions despite the agreement in the ozone–temperature

trends, the ozone mixing ratios are consistently overestimated in all temperature bins. Since the IP and EA are the only regions where the model overestimates the ozone mixing ratios in both the first two temperature bins (< 25˚C), this might imply an overestimation in the background ozone levels which might partially mask some of the underestimation of ozone mixing ratios in the last two temperature bins (≥ 25˚C). On the other hand, for the BX region a combined emissions increase scenario ($2NOx$,1.5-2VOC) is required. Finally, for the BI and SC regions, the base case performs quite well.

### 3.3 Sensitivity of ozone to meteorology

Meteorology affects ozone mixing ratios not only directly (e.g. horizontal advection, vertical diffusion, photolysis rates, etc.) but also indirectly by influencing the concentrations of its precursors and its chemistry. Therefore, we performed some tests to explore the impact of key meteorological parameters like temperature and wind speed. The PBLH is another meteorological parameter that can have a strong influence on ozone mixing ratios, but its impact is very complex and can have opposite effects: Increased vertical mixing dilutes the ozone precursors inhibiting ozone production, but it also reduces the $NO_x$ titration of ozone (especially in urban areas) and enhances the downward transport of ozone from the enriched-ozone upper layers in the evening-morning which in turn influences the ozone mixing ratios and chemistry the next day (Kleeman, 2008; Lin et al., 2008). It is probably due to these reasons that the correlation of ozone mixing ratios with the mixing depth is reported to be weak (Wise and Comrie, 2005; Ordóñez et al., 2005; Jacob and Winner, 2009). Therefore, we do not expect that our PBLH uncertainties (Table 4, Fig. S8) could consistently explain the observed bias trend in the afternoon ozone mixing ratios and since there is no straightforward and consistent way to artificially perturb the PBLH, we did not perform such a sensitivity test. The results of the ozone sensitivity to the tested meteorological parameters are shown in Fig. 11.

**Temperature:** A temperature increase of 4˚C was chosen to be tested as the model underestimates the observed high temperatures ($\geq$ 25˚C) in most of the domain by ~1–2˚C (Fig. S4) and by ~3–4˚C in the PV region (interpolation errors are higher for coastal and mountain areas). As expected, increasing the temperature by 4˚C (green color) causes an increase in ozone mixing ratios in the range of 1.5–6 ppb for all regions (Fig. 11). The main driver of enhanced ozone production (excluding temperature driven emission changes) due to a temperature increase is peroxyacetyl nitrate (PAN) (including similar compounds) chemistry (Baertsch-Ritter et al., 2004; Dawson et al., 2007; Pusede et al., 2015). Those studies explain that PAN can serve as $NO_x$ and radical reservoir and redistribute them away from the large emission areas (e.g. cities, power plants) to more remote, rural ones (by thermally decomposing back to $NO_2$ and radicals). A temperature increase will shift the equilibrium between $NO_2$ and PAN to higher $NO_2$ mixing ratios and thus enhance ozone production. However, as mentioned earlier, the true impact of the temperature uncertainty in our simulations is lower than the tested one, as our meteorological model evaluation indicates a good prediction of the surface temperature for most of the stations (Table 4, Fig. S4 and S6). Moreover, at the higher altitudes the prediction of temperature is also good with the afternoon MB being within about a $\pm$ 1 ˚C range for most stations (Fig. S9). Since for the given temperature uncertainty of our meteorological input the impact on the ozone mixing ratio is much less, this cannot explain the magnitude and trend of the bias seen in Fig. 11.

**Wind speed:** The processes that are mainly influenced by a change in wind speed are: advection, horizontal diffusion and dry deposition. As a result of this complex effect, the impact of the wind speed on ozone mixing ratios (blue and purple colors) is less systematic than that of the temperature with its correlation with the region's air pollution. More specifically, in the less polluted Scandinavia (SC region) the effect of the wind speed reduction is lower in the highest bins ($\geq$ 60 ppb) compared to the rest of the bins. This is possible due to the fact that the low mixing ratios of ozone precursors in the region

do not lead to a significant accumulation when the wind speed is reduced and hence ozone production does not increase as much as in the other regions. On the other hand, when we double the wind speed (purple color) the ozone precursors are rapidly driven away inhibiting any ozone production and the wind speed effect in SC region increases with increasing ozone bin. Indeed, this model sensitivity pattern is also observed for rest of the regions and for both wind speed tests (WS/2 and

WSx2 scenarios), as there is an increase with increasing bin in the ozone enhancement (reduction) by the wind speed reduction (increase), from approximately 5 to 11 (-2 to -10) ppb for the IP, MD, PV, EA and ME regions, from 6 to 11 (-2 to -11) ppb for the BX region and from 1 to 11 (-1 to -12) ppb for the BI region. In general, the actual impact of the wind speed bias on ozone mixing ratios will be lower (for both wind speed sensitivity scenarios), as the model provides a quite good prediction of wind speed (as shown in Section 3.1) with the majority of the stations having an afternoon (12:00–18:00 UTC)

mean bias within a range of $\pm$ 1 m s$^{-1}$ (Fig. S7). Moreover, the increase of the wind speed seems more representative for our case, since the mean bias of the wind speed for observed wind speeds $\geq$ 2 m s$^{-1}$ is negative (except for the SC region) in the range of a factor of 1.5–2 for the majority of the samples (Fig. S5). The model only consistently overestimates the very low wind speeds (0–2 m s$^{-1}$) with the mean bias ranging, depending on the region, from 0.5 to 1.5 m s$^{-1}$. However, these low wind speeds comprise less than 20% of the total sample (Fig. S5) for most regions, with the exception of the MD, EA and

PV regions where it is higher (20%, 22% and 32%, respectively). Regarding the wind speed at the higher altitudes, the vertical wind speed profiles indicate a mean bias of $\pm$ 1 m s$^{-1}$ for most heights (Fig. S10), which is rather small for the high wind speeds of the higher levels of the atmosphere. Consequently, the tested absolute decrease (increase) of wind speed is higher at higher altitudes and hence the dilution of ozone precursors is even lower (higher) than near the surface. This results in enhanced (reduced) ozone production within about the first kilometer from the surface, where photochemistry can be

responsible for about a third of the ozone mixing ratio variability (Chevalier et al., 2007), leading usually to higher (lower) ozone mixing ratios at these altitudes than the respective ones in the base case (Fig. S11). The reduced (increased) dry deposition also accounts for the increased (reduced) ozone mixing ratios near the surface. In general, reducing the wind speed by half does result in an improvement of the underestimation of high ozone mixing ratios, but at the same time worsens significantly the overestimation of the low ozone mixing ratios. On the other hand, the sign of the wind speed bias is

in most cases and in most regions negative, justifying more the WSx2 scenario (compared to the WS/2 one) which will improve the model performance in the lower ozone bins but also unmask a higher underestimation in the higher ozone bins. In addition, the wind speed uncertainties that were tested here are much higher than the ones from the model evaluation results (especially for the upper layers). Therefore, we conclude that the uncertainty in wind speed cannot be the reason for the bias in the afternoon ozone mixing ratios.

**Ozone–temperature correlation:** A closer look to the influence of meteorology on the ozone production is shown in Fig. 12, with the use of the ozone–temperature correlation. Since the impact of meteorology on ozone mixing ratios was thoroughly examined in Fig. 11, the focus of Fig. 12 is more on the effect of meteorology on the correlation of ozone with temperature. It has to be noted that the ozone mixing ratio for the T+4°C scenario (green) is plotted against the temperature of the base case. The use of the temperature range (bin) instead of a single value makes the ozone–temperature regression

less sensitive to uncertainties related to the temperature bias (within the acceptable margins of an evaluated meteorological model performance). Any over- or under-estimation in ozone mixing ratios due to temperature bias will be averaged out if they are in the same temperature bin. Even if the ozone mixing ratios are wrongly allocated in a different bin (due to the temperature bias), this won't affect the overall ozone–temperature regression as these biased ozone mixing ratios will be in

the same range with correctly predicted ozone mixing ratios for the same temperature bin. In other words, if the T+4˚C scenario is plotted consistently in Fig. 12, then the impact of the temperature on ozone becomes really small ($\leq 1$ ppb) in all temperature bins and for all regions. Since a wind speed reduction (blue) and increase (purple) consistently increases and decreases the ozone mixing ratios (both low and high values), respectively, this leads to negligible changes in the ozone–temperature trend. Especially for the MD, PV, EA, ME and BX regions, the lines of WS/2 and WSx2 scenarios are almost

parallel to the base case (red) which is much less steep than the observed one (black).

Overall, the meteorological scenarios that were tested did not improve the modeled ozone performance as consistently as some of the emissions scenarios. The behavior of the modeled wind speed biases, i.e. overestimation of the lowest wind speed and underestimation of the rest (Fig. S5), can explain to some degree the overestimation of low ozone mixing ratios and the underestimation of the high ones; but not entirely since the tested wind speed uncertainties are higher than the real

ones which are indicated by the model performance evaluation (Table 4, Fig. S5, S7, S10). The temperature sensitivity test had a smaller impact than the one of the wind speed, and also the tested change (+ 4˚C) was higher than the actual model temperature bias range ($\pm 2$˚C) for most of the parts of the domain. In general, the meteorology does not seem to be the main source of error for the underestimation of ozone production, in contrast to the emissions.

### 3.4 Sensitivity of ozone to initial and boundary conditions

Many studies (Katragkou et al., 2010; Solazzo et al., 2013; Giordano et al., 2015; Im et al., 2015) have reported a strong influence of the boundary conditions on ozone mixing ratios, but their impact is less significant near the surface and inside the PBL, as well as in summer compared to other seasons (winter or autumn), due to more dominant near-surface effects (e.g. photochemistry, emissions, transport, dry deposition). Furthermore, Katragkou et al. (2010) showed that the impact of increased $O_3$ in the lateral boundaries by 8 ppb in Europe in summer was already down to half (3–4 ppb) over Great Britain

and western Scandinavia and faded out towards central and southeast Europe. In addition, an increase of 12 ppb of $O_3$ in the top boundary and 1 ppb of $NO_x$ in the lateral boundaries resulted in less than about 2 and 3 ppb increase, respectively, in surface ozone over whole Europe. In order to investigate the influence of background ozone levels on surface ozone mixing ratios, we perturbed the initial and boundary (lateral and top) conditions (ICBC) of ozone by $\pm 5$ ppb (Table 3). The impact of an increase (decrease) in the ICBC of ozone was a 1–2 ppb increase (decrease) consistently in all ozone bins and in most

regions (see Fig. S12). The tested impact on surface ozone diminished as it progressed into the interior of the domain (not shown), which is in line with the aforementioned results reported in the literature. Therefore, uncertainties in the ICBC do not seem to be responsible for the observed ozone bias trend in the surface mixing ratios.

## 4 Conclusions

In this work we used alternative methods to evaluate the modeled surface afternoon ozone mixing ratios and production more consistently in the whole European domain for the summer of 2010 using the regional air quality model CAMx. The results were analyzed in eight European regions. The separation of the observed surface ozone mixing ratios in bins helps to

unmask the hidden model bias and identify the significant underestimation of high mixing ratios and overestimation of the low ones. Since the high surface ozone mixing ratios are more related to photochemical ozone production, an evaluation of the modeled ozone production was carried out using the ozone–temperature correlation. The use of the modeled temperature as a surrogate for the observed one (after the validation of this hypothesis) allowed us to perform the modeled ozone production evaluation for most of the stations in the whole European domain. As an additional, alternative approach to the

ozone–temperature correlation, we divided the modeled temperature into bins and paired it to the respective observed and modeled surface ozone mixing ratios. The results indicated that the modeled surface ozone mixing ratios have a less steep increase with temperature than the observed ones. The modeled ozone–temperature regression slope (ppb $°C^{-1}$) is underestimated by about a factor of 2 for most stations. In addition, the use of the relationship between ozone and temperature bins showed the model underestimation of both high ozone mixing ratios and ozone–temperature trend. In order

to characterize the sources of uncertainty that led to the aforementioned model behavior, model sensitivity tests were performed to investigate the influence of emissions, meteorology and initial and boundary conditions.

Increasing just the VOC emissions by a factor of 1.5 and 2 for the anthropogenic and biogenic emissions, respectively, resulted in a small increase of surface ozone mixing ratios (1–2 ppb) across all observed ozone mixing ratio bins for most of the regions but the Mediterranean (MD), Po Valley (PV) and Benelux (BX) regions where the impact was higher (2–4 ppb).

On the contrary, the doubling of only the $NO_x$ emissions resulted in a more significant increase of ozone (6–8 ppb) in the higher observed ozone mixing ratio bins in all regions apart from the BX region where it slightly decreased. The effect in the lowest observed ozone mixing ratio bins was either an increase or decrease of ozone depending on the region due to enhanced $NO_x$ titration. The combined increase of $NO_x$ and VOC emissions increased the ozone mixing ratios even more in all bins and regions except for the lower ozone bins in the BX region where the ozone mixing ratio decreased. Overall, the

best model performance improvement was brought by the increase of $NO_x$ emissions for southern (Iberian Peninsula (IP) and Mediterranean (MD) regions), central (Po Valley (PV) and Mid-Europe (ME) regions) and eastern Europe (EA region), and by the combined increase of $NO_x$ and VOC emissions for the Benelux area (BX region). Increasing only traffic $NO_x$ emissions by a factor of 4 had almost the same impact as doubling all $NO_x$ emissions. However, as discussed in Sect. 2.3, previous investigations indicate higher uncertainties in $NO_x$ emissions from the road-transport compared to other sectors.

Therefore, the 4traf_$NO_x$ scenario is more consistent with the previous studies than the 2$NO_x$ scenario, suggesting that high uncertainties in the $NO_x$ emissions from road-transport are more likely to be the main reason for underestimated ozone production rather than uncertainties in emissions from other sectors. For the less polluted British Isles (BI) and Scandinavia

(SC) regions no emission adjustment was necessary. The evaluation of ozone–temperature correlation for these emission scenarios also led to the same conclusions.

Both sensitivity tests with increased temperatures by 4˚C and with the reduced wind speed by 50% led to a significant increase (1.5–6 and 7–10 ppb, respectively) in surface ozone mixing ratios in all mixing ratio bins and regions except for the SC region where the impact of wind speed reduction was less. On the contrary, the doubling of the wind speed led to a more significant decrease (-6 to -12 ppb) in surface ozone mixing ratios in the higher bins for all regions, but the impact decreased with decreasing bin ranging from 0 to -6 ppb, depending on the region. Although the T+4˚C and WS/2 scenarios might have improved the underestimation of the observed high ozone mixing ratios, they significantly enhanced the overestimation of the respective low ones and vice versa for the WSx2 scenario. The same conclusions were reached by the evaluation of the ozone–temperature correlation for these tests. In addition, the tested meteorological perturbations were much higher than the uncertainties in this study and therefore their impact on ozone is expected to be lower. Additional tests with perturbed initial and boundary conditions showed a small effect consistently in all mixing ratio bins and regions.

The results obtained in this study indicate that the uncertainties in emissions (especially the too low traffic $NO_x$ emissions in the inventories) are mainly responsible for the underestimation of the observed high summer ozone mixing ratios and ozone production in Europe. These uncertainties also seemed to vary spatially, since different regions had different responses to the same tested emission changes. Further investigation of the emission uncertainties and improvement of the modeled ozone production will contribute to more consistent and effective ozone mitigation strategies for the future.

*Data availability*. All data are available upon request from the corresponding authors. References to the repositories of the observational data used have been also provided in Sect. 2.

*Competing interests*. The authors declare that they have no conflict of interest.

*Acknowledgments*. We would like to thank the following agencies for preparing the datasets used in this study: TNO for the anthropogenic emission inventory; the European Environmental Agency (EEA) for the air quality data; the European Centre for Medium-Range Weather Forecasts (ECMWF) and British Atmospheric Data Centre (BADC) for the meteorological data; the United States Geological Survey (USGS) for the land-use data; the World Ozone and Ultraviolet Data Centre (WOUDC) and its data-contributing agencies for the ozonesonde profiles; the University of Wyoming for the radiosonde profiles; the National Aeronautics and Space Administration (NASA) and its data-contributing agencies (NCAR, UCAR) for the TOMS and MODIS data, the global air quality model data and the TUV model. Calculations of meteorological data were performed with the Swiss National Supercomputing Centre (CSCS). Our thanks extend to RAMBOLL ENVIRON and especially Cristopher Emery for their continuous support of the CAMx model. This work was financially supported by the Swiss Federal Office of Environment (FOEN).

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

**Tables**

10   **Table 1.** Definition of statistical metrics for model performance evaluation. $M_i$ and $O_i$ stand for modeled and observed values respectively and $N$ being the total number of paired values.

| Metric | Definition |
|---|---|
| Mean Bias (MB) | $MB = \dfrac{1}{N}\sum_{i=1}^{N}(M_i - O_i)$ |
| Mean Gross Error (MGE) | $MGE = \dfrac{1}{N}\sum_{i=1}^{N}|M_i - O_i|$ |
| Root-Mean-Square Error (RMSE) | $RMSE = \sqrt{\dfrac{1}{N}\sum_{i=1}^{N}(M_i - O_i)^2}$ |
| Index of Agreement (IOA) | $IOA = 1 - \dfrac{N \cdot RMSE^2}{\sum_{i=1}^{N}(|M_i - \bar{O}| + |O_i - \bar{O}|)^2}$ |
| Pearson correlation coefficient ($r$) | $r = \dfrac{\sum_{i=1}^{N}(M_i - \bar{M}) \cdot (O_i - \bar{O})}{\sqrt{\sum_{i=1}^{N}(M_i - \bar{M})^2} \cdot \sqrt{\sum_{i=1}^{N}(O_i - \bar{O})^2}}$ |
| Mean Fractional Bias (MFB) | $MFB = \dfrac{1}{N}\sum_{i=1}^{N}\dfrac{2 \cdot (M_i - O_i)}{M_i + O_i}$ |
| Mean Fractional Error (MFE) | $MFE = \dfrac{1}{N}\sum_{i=1}^{N}\dfrac{2 \cdot |M_i - O_i|}{M_i + O_i}$ |

5 **Table 2.** Performance criteria and goals for model results (from Emery et al., 2001; EPA, 2007; Boylan and Russel, 2006).

| Parameter | Metric | Criteria | Goal |
|---|---|---|---|
| Temperature (T) | MB | $\leq \pm 0.5$ K | |
| | MGE | $\leq 2$ K | – |
| | IOA | $\geq 0.8$ | |
| Wind Speed (WS) | MB | $\leq \pm 0.5$ m s$^{-1}$ | |
| | RMSE | $\leq 2$ m s$^{-1}$ | – |
| | IOA | $\geq 0.6$ | |
| Wind Direction (WD) | MB | $\leq \pm 10$ deg | |
| | MGE | $\leq 30$ deg | – |
| Humidity (expressed as water vapor mixing ratio ($q_v$)) | MB | $\leq \pm 1$ g/kg | |
| | MGE | $\leq 2$ g/kg | – |
| | IOA | $\geq 0.6$ | |
| PM$_{2.5}$ | MFB | $\leq \pm 60\%$ | $\leq \pm 30\%$ |
| | MFE | $\leq 75\%$ | $\leq 50\%$ |
| O$_3$ | MFB | $\leq \pm 30\%$ | $\leq \pm 15\%$ |
| | MFE | $\leq 45\%$ | $\leq 30\%$ |

**Table 3.** Description of sensitivity tests.

| Scenario | Description |
|---|---|
| Base | Base case using the meteorological and emission data as described in Sect. 2.2 and 2.3, respectively |
| 1.5-2VOC | Increased VOC emissions: by a factor of 1.5 and 2 for the anthropogenic and biogenic VOC, respectively. |
| $2NO_x$ | Increased $NO_x$ emissions by a factor of 2. |
| 1.5-2VOC,$2NO_x$ | Combination of scenarios 1.5-2VOC and $2NO_x$. |
| 4traf_$NO_x$ | Increased $NO_x$ emissions only in the road-transport sector (SNAP 7) by a factor of 4. |
| T+4˚C | Increased first layer air temperature by 4˚C. Impact on emissions was excluded. |
| WS/2 | Reduced horizontal wind speed at all altitudes by 50%. Vertical wind speed is calculated inside CAMx to be consistent with the continuity equation and ensure mass conservation. |
| WSx2 | Increased horizontal wind speed at all altitudes by a factor of 2. Vertical wind speed is calculated inside CAMx to be consistent with the continuity equation and ensure mass conservation. |
| $\pm5O_3$ | Increased/decreased initial and boundary (top and lateral) conditions of ozone by 5 ppb. |

**Table 4.** Model performance evaluation for the meteorological parameters in summer (JJA) 2010.

| | MB | MGE | RMSE | IOA (–) | $r$ (–) |
|---|---|---|---|---|---|
| T (˚C) | -0.5 | 1.7 | 2.1 | 0.9 | 0.9 |
| WS (m s$^{-1}$) | -0.2 | 1.5 | 2.0 | 0.6 | 0.5 |
| WD (deg) | 10.0 | 35 | – | – | – |
| $q_v$ (g kg$^{-1}$) | 0.02 | 1.0 | 1.3 | 0.9 | 0.8 |
| PBLH (m) | 45 | 370 | 485 | 0.7 | 0.5 |

**Table 5.** Model performance evaluation for the daily mean concentrations of the chemical species in summer (JJA) 2010. The units for MB, MGE and RMSE are in ppb for the gas species and in $\mu$g m$^{-3}$ for the PM$_{2.5}$.

| | No. of stations | MB | MGE | RMSE | MFB (%) | MFE (%) | $r$ (–) |
|---|---|---|---|---|---|---|---|
| O$_3$ | 347 | 4 | 7 | 8 | 12 | 20 | 0.7 |
| NO$_2$ | 228 | -0.2 | 2 | 3 | -17 | 53 | 0.4 |
| SO$_2$ | 107 | 1 | 2 | 2 | 42 | 81 | 0.3 |
| CO | 27 | -72 | 77 | 89 | -41 | 47 | 0.2 |
| PM$_{2.5}$ | 23 | -0.4 | 5 | 7 | -2 | 41 | 0.5 |

**Figures**

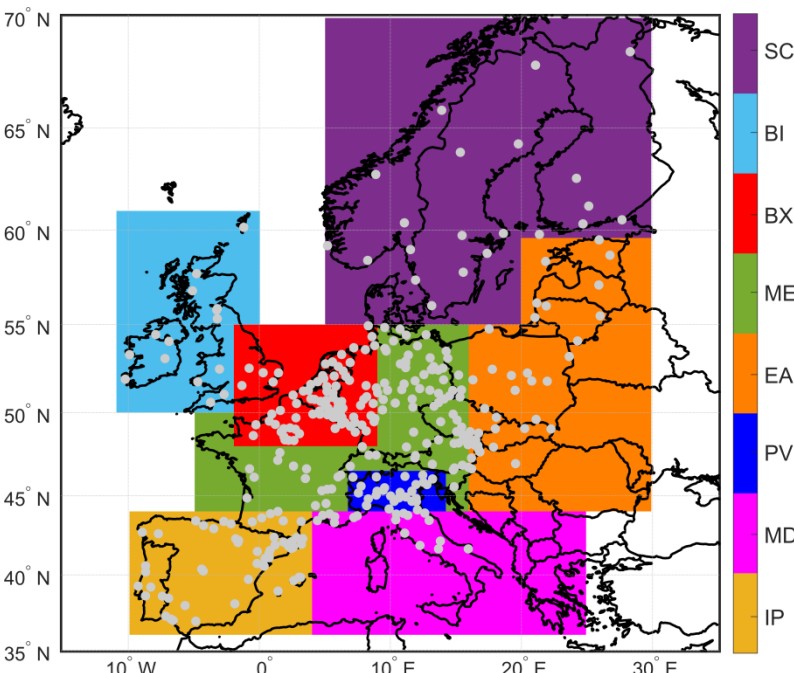

**Figure 1.** The European model domain and its sub-regions: Iberian Peninsula (IP), Mediterranean (MD), Po Valley (PV), Eastern Europe (EA), Mid-Europe (ME), Benelux (BX), British Isles (BI) and Scandinavia (SC). Grey dots indicate the rural background Airbase stations of the hourly ozone measurements.

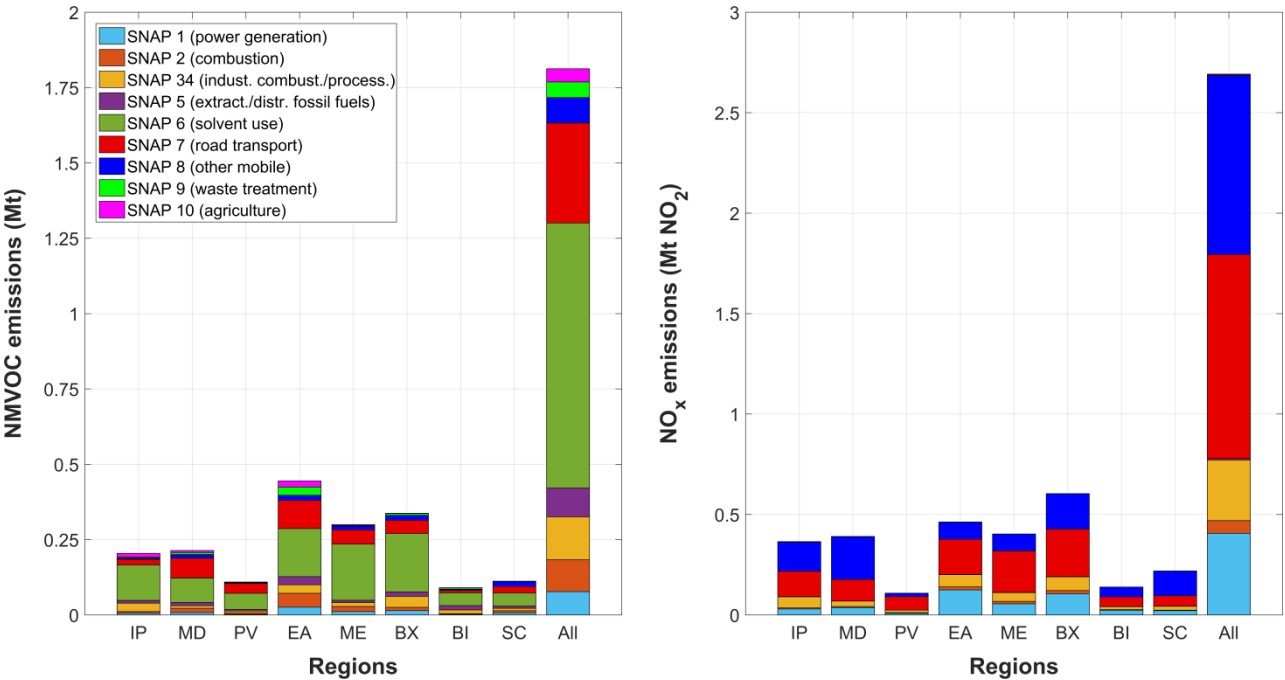

**Figure 2.** Total NMVOC (left panel) and NO$_x$ (right panel) emissions per SNAP category for each region in Europe as well as for their sum for summer 2010. A detailed description of the SNAP source categories is given in Table S2.

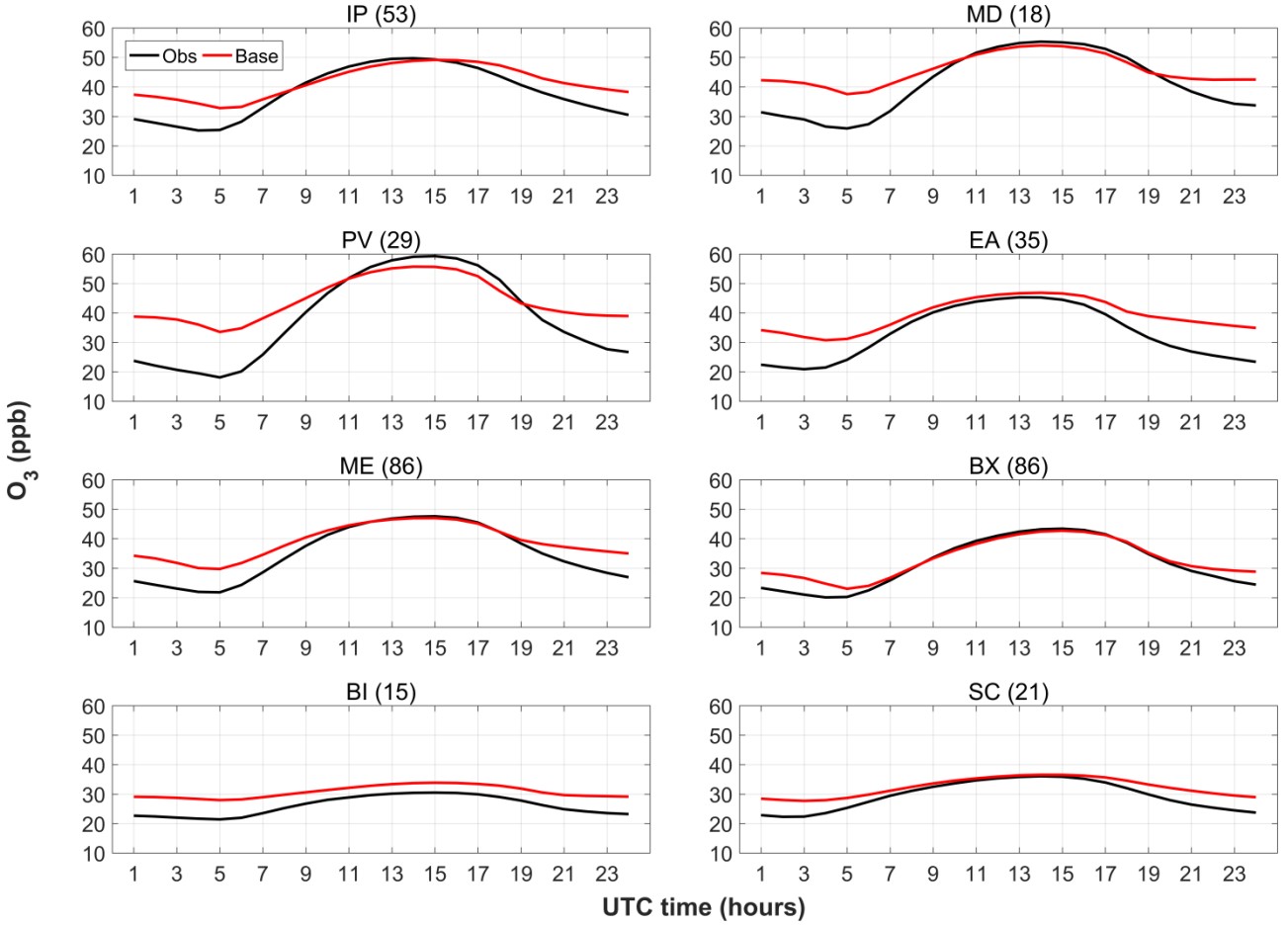

**Figure 3.** Diurnal profiles of surface O₃ mixing ratios in 8 European regions in summer 2010. The number of stations available for each region is reported in parentheses at the top of each panel.

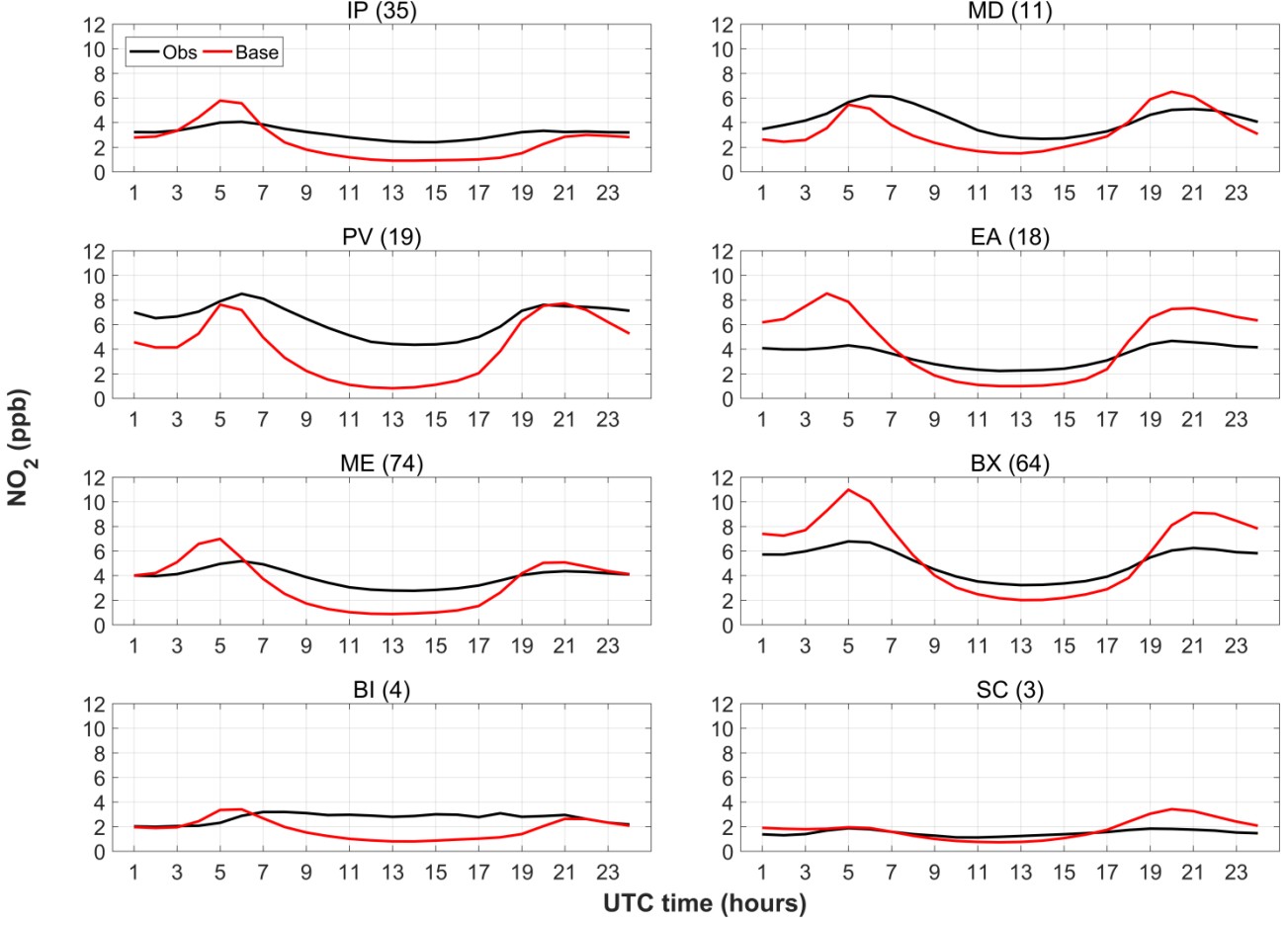

**Figure 4.** Diurnal profiles of the surface NO$_2$ mixing ratios in 8 European regions in summer 2010.The number of stations available for each region is reported in parentheses at the top of each panel.

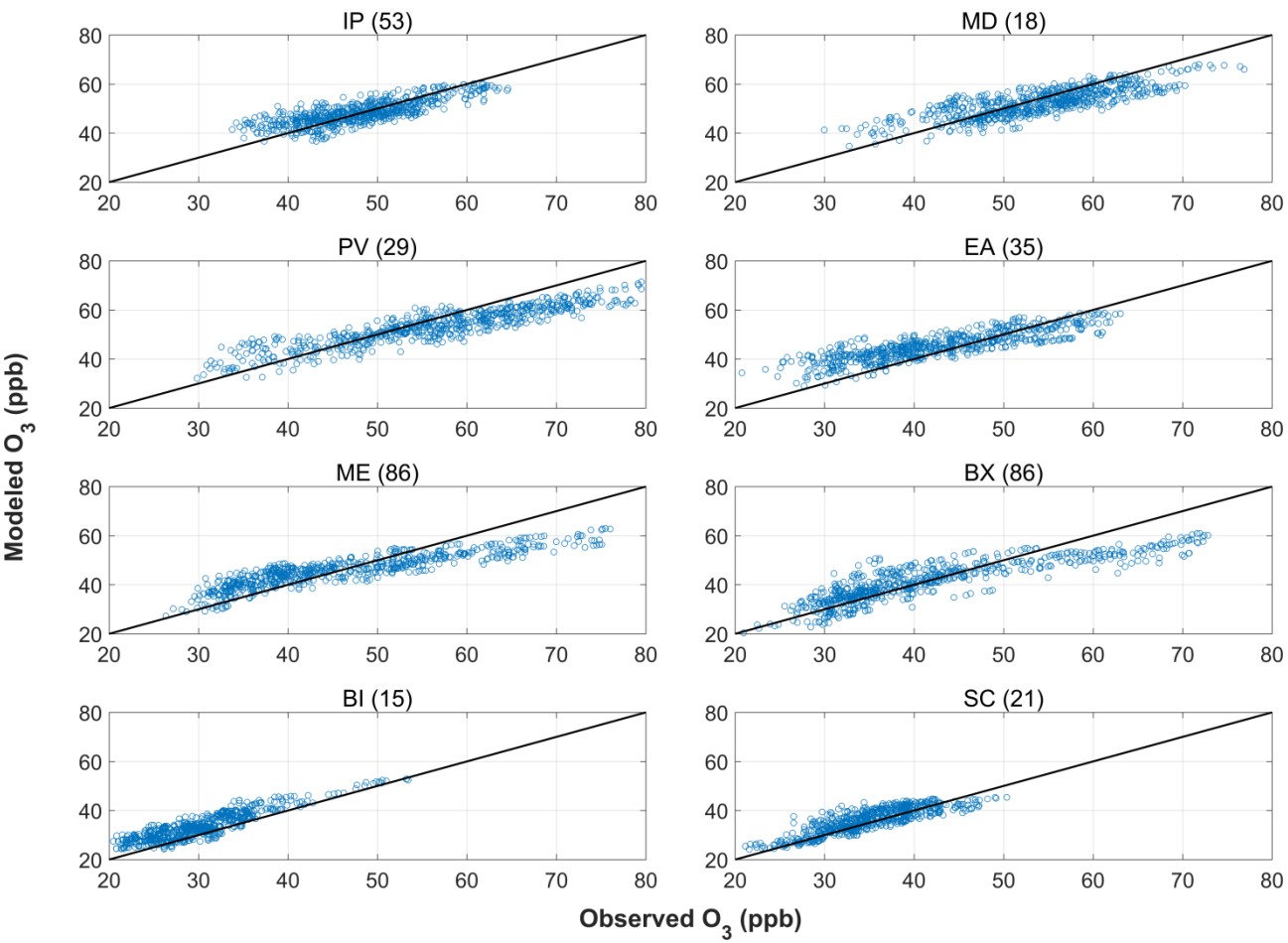

**Figure 5.** Scatterplots of modeled vs. observed surface afternoon (12:00–18:00 UTC) mean $O_3$ mixing ratios in 8 European regions in summer 2010. The number of stations available for each region is reported in parentheses at the top of each panel.

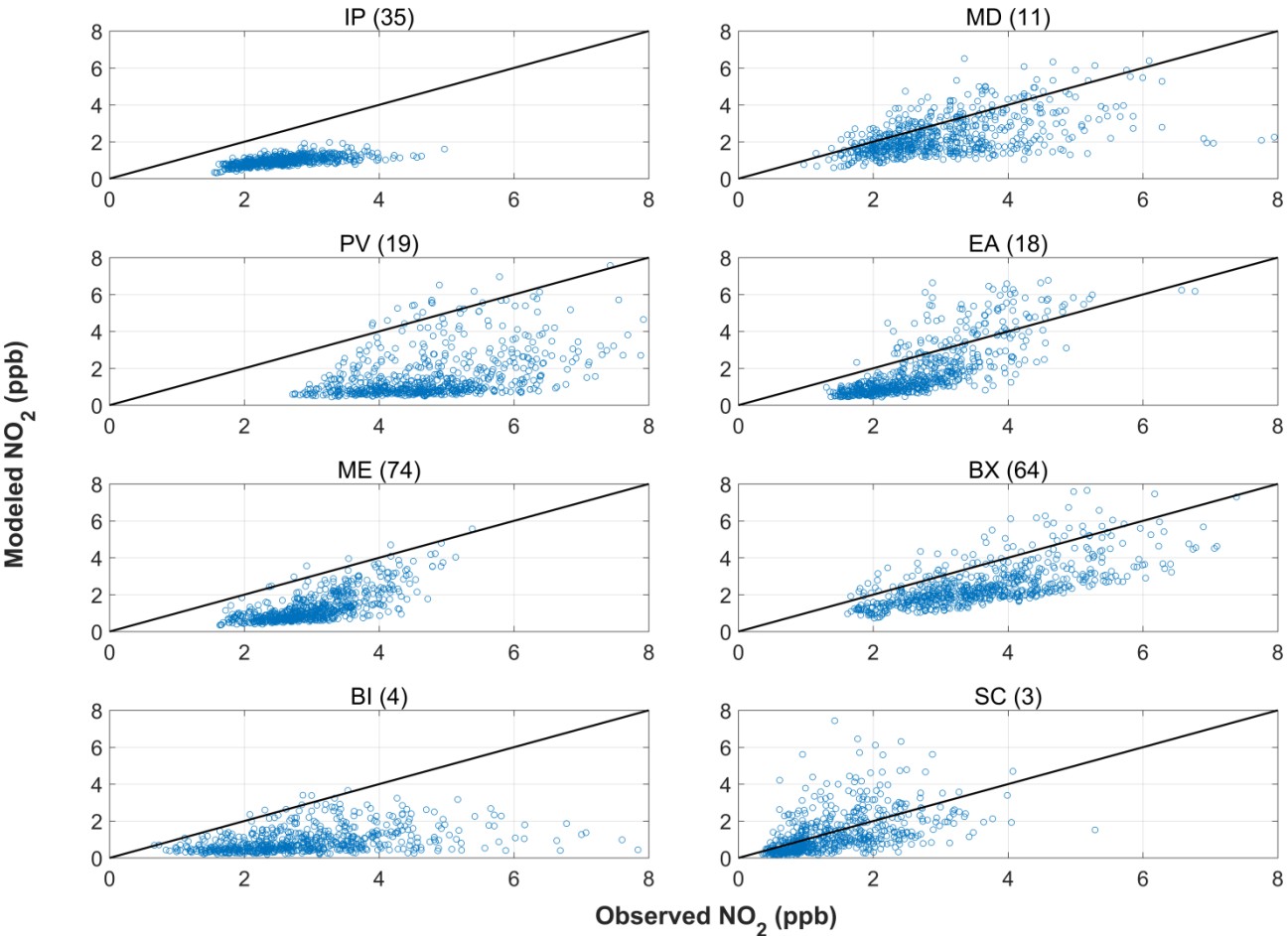

**Figure 6.** Scatterplots of modeled vs. observed surface afternoon (12:00–18:00 UTC) mean $NO_2$ mixing ratios in 8 European regions in summer 2010. The number of stations available for each region is reported in parentheses at the top of each panel.

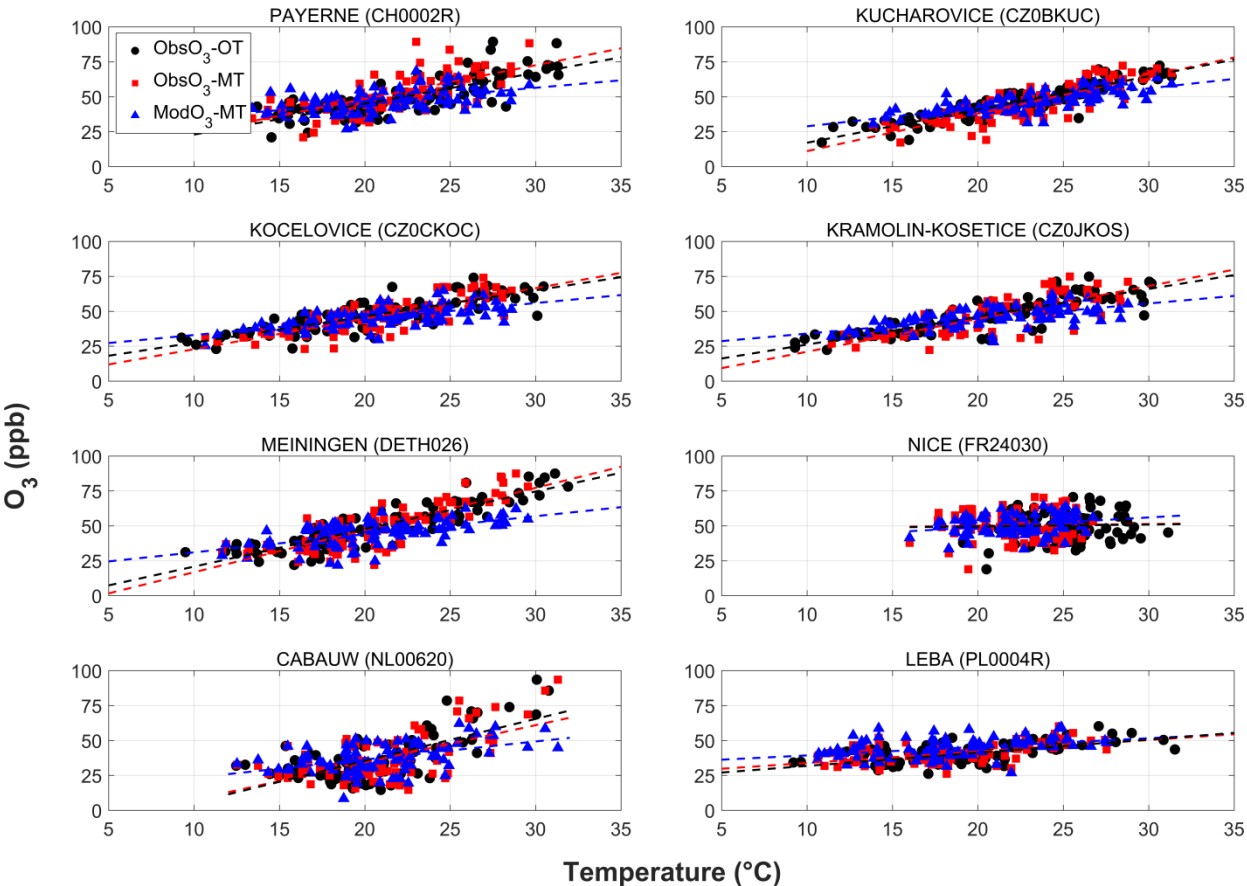

**Figure 7.** Scatterplots of surface afternoon (12:00–18:00 UTC) mean $O_3$ mixing ratios vs. temperature for 8 stations in summer 2010. Observed $O_3$ mixing ratios are plotted against both observed (ObsO$_3$-OT) and modeled (ObsO$_3$-MT) temperature, while the modeled $O_3$ mixing ratios are plotted only against the modeled temperature (ModO$_3$-MT). Dashed colored lines represent the best linear fit for each case.

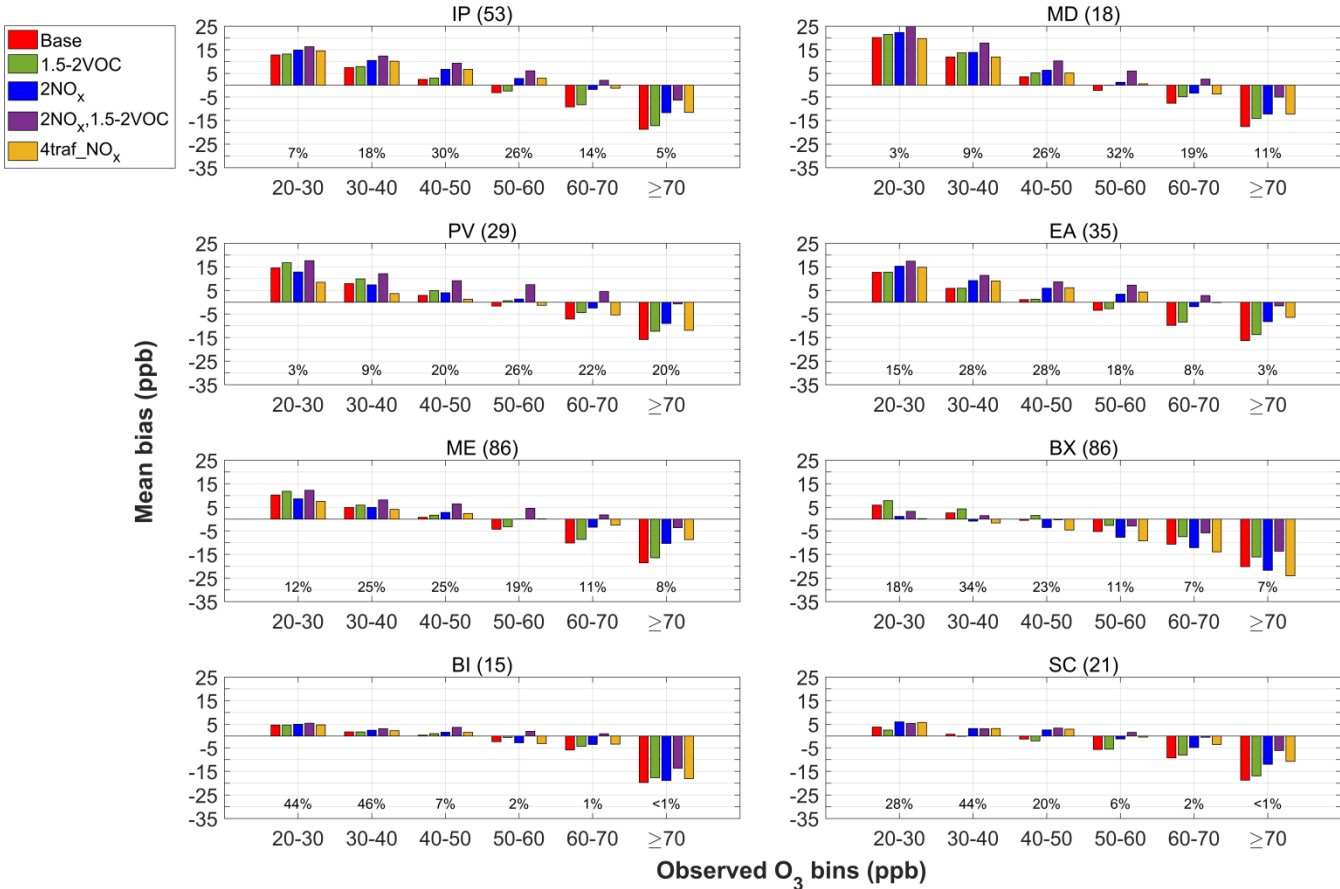

**Figure 8.** Mean bias of the afternoon (12:00–18:00 UTC) surface $O_3$ mixing ratios for each bin of observed surface $O_3$ mixing ratios for various emissions scenarios in 8 European regions in summer 2010. Percentage values below the bars indicate the fraction of the values assigned to each bin for each region. The number of stations available for each region is reported in parentheses at the top of each panel.

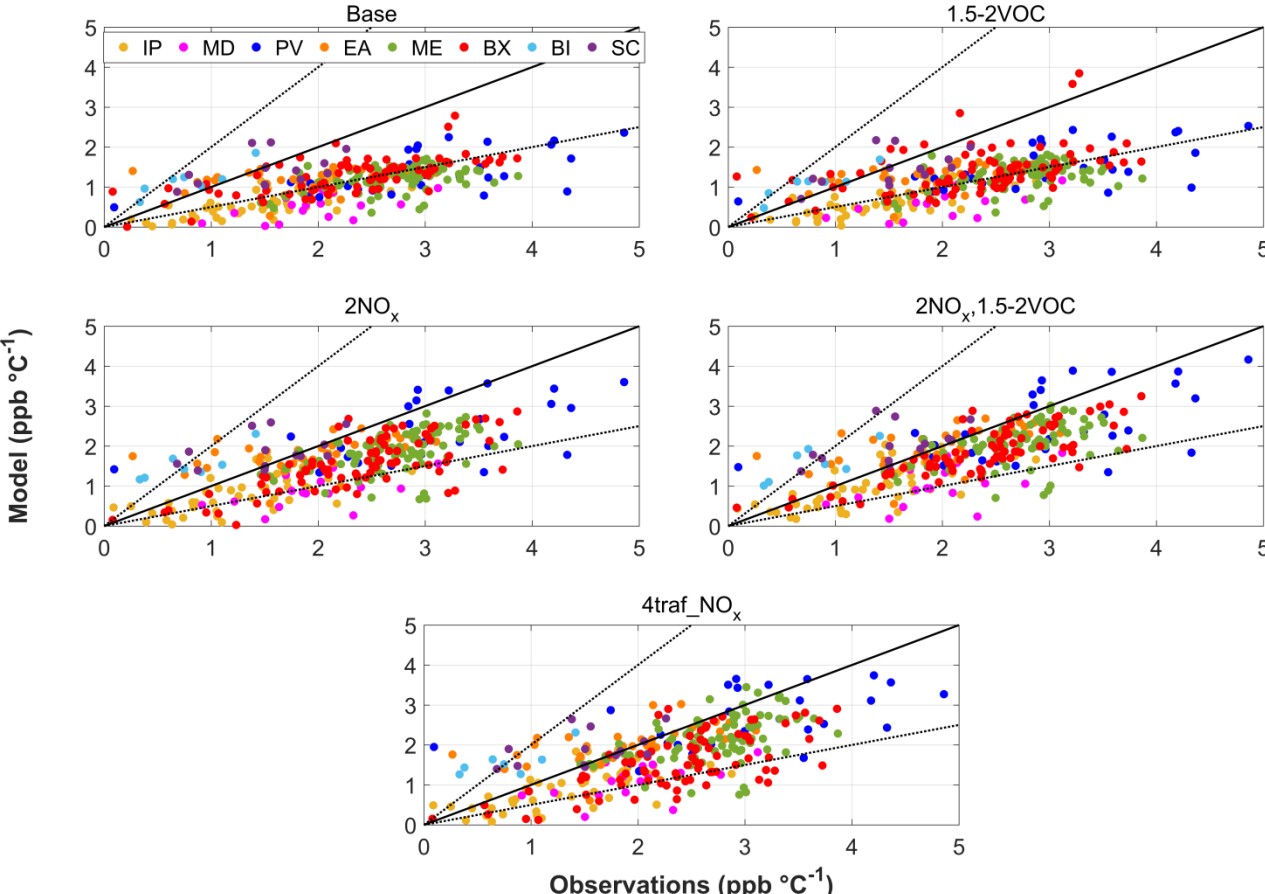

**Figure 9.** Scatterplots of the modeled versus observed surface afternoon (12:00–18:00 UTC) mean $O_3$ – temperature linear regression slope for each station for various emission scenarios in 8 European regions in summer 2010. The solid black line is the 1:1 line and the dotted black lines are the 2:1 and 1:2 lines.

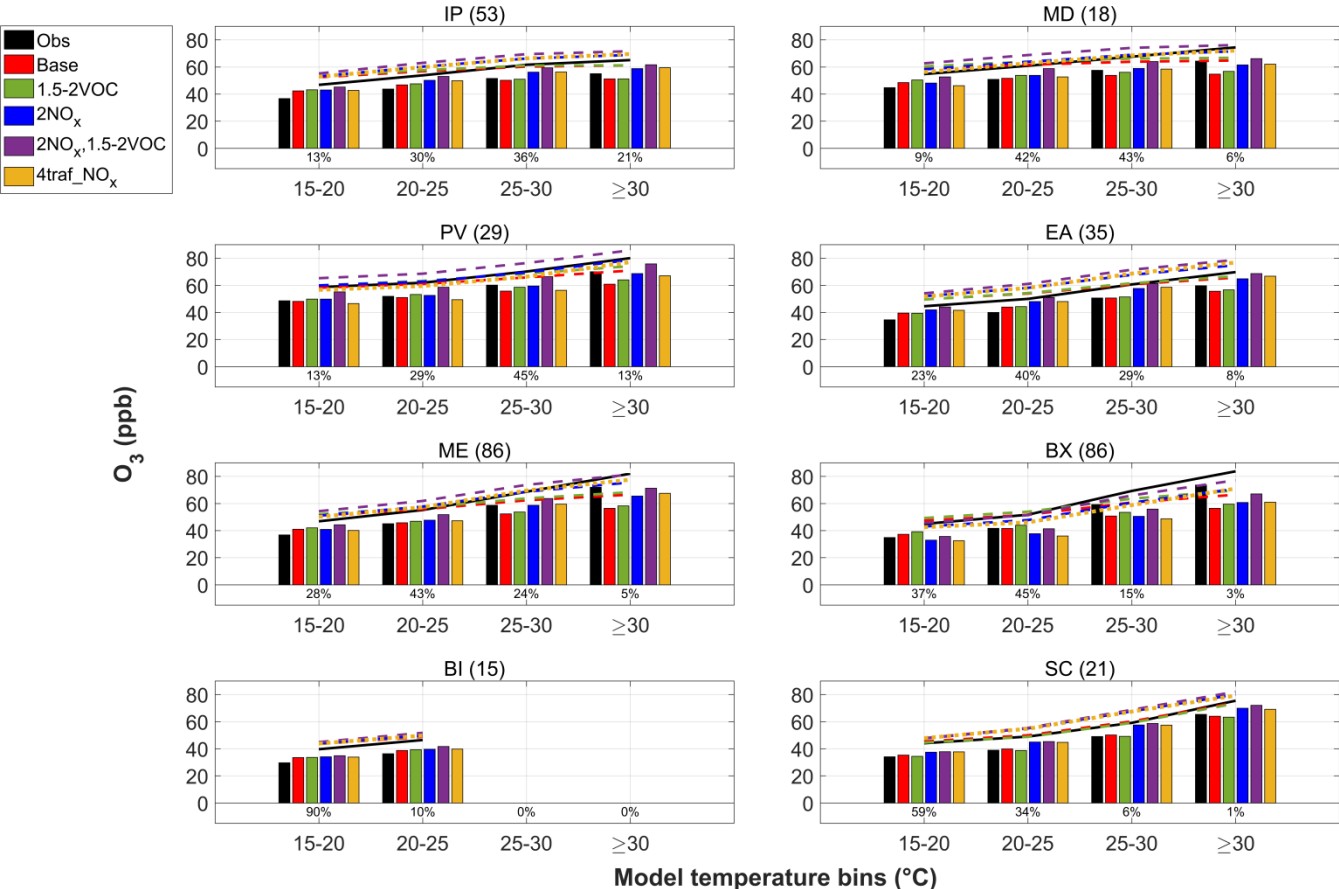

**Figure 10.** Afternoon (12:00–18:00 UTC) surface $O_3$ mixing ratios for each modeled temperature bin for various emissions scenarios in 8 European regions in summer 2010. Colored lines show the trends of the respective bars and are shifted up by 10 ppb for visualization purposes. Percentage values below the bars indicate the fraction of the values assigned to each bin for each region. The number of stations available for each region is reported in parentheses at the top of each panel.

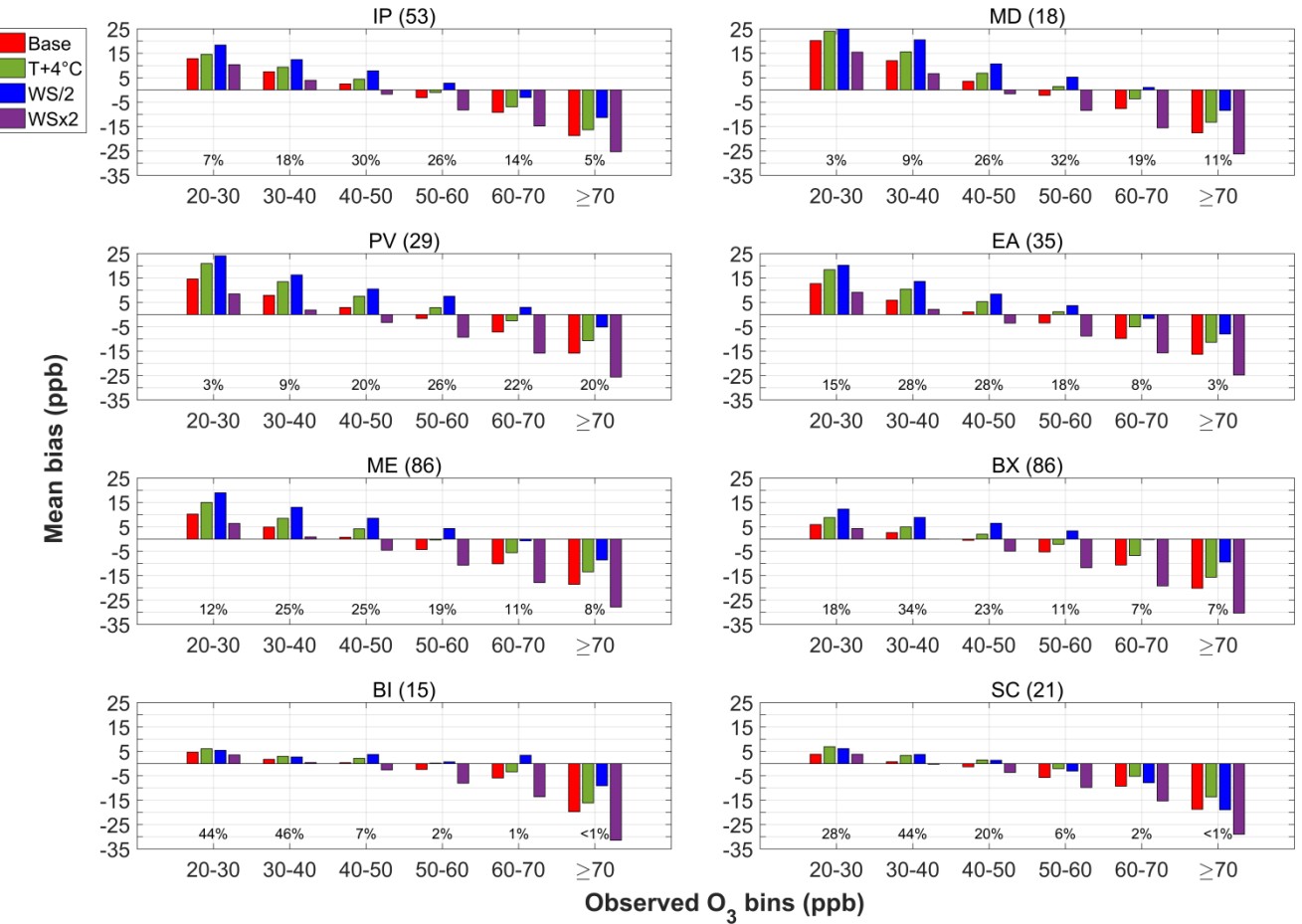

**Figure 11.** Mean bias of the afternoon (12:00–18:00 UTC) surface $O_3$ mixing ratios for each bin of observed surface $O_3$ mixing ratios for various meteorological scenarios in 8 European regions in summer 2010. Percentage values below the bars indicate the fraction of the values assigned to each bin for each region. The number of stations available for each region is reported in parentheses at the top of each panel.

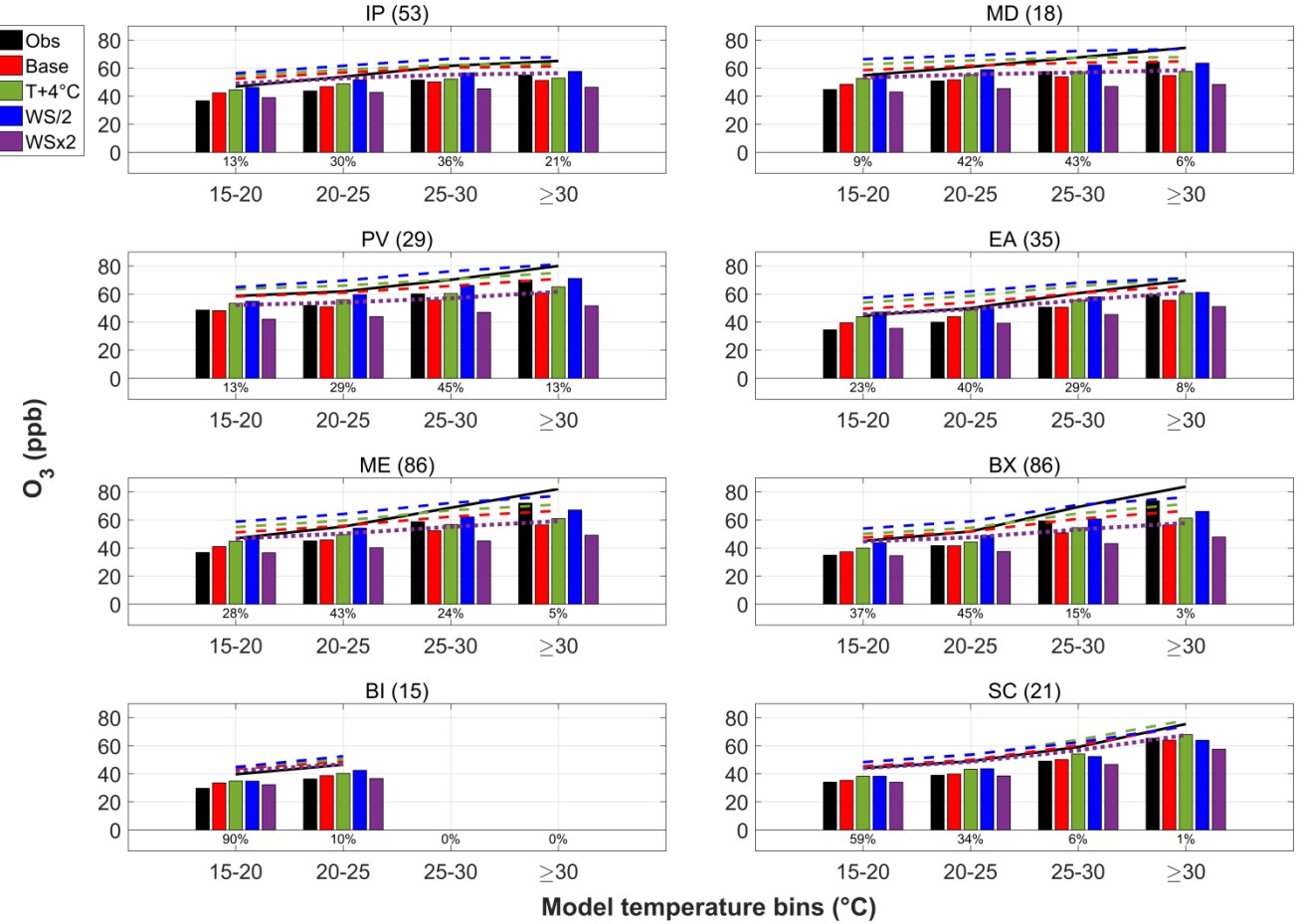

**Figure 12.** Afternoon (12:00–18:00 UTC) surface $O_3$ mixing ratios for each modeled temperature bin for various meteorological scenarios in 8 European regions in summer 2010. Colored lines show the trends of the respective bars and are shifted up by 10 ppb for visualization purposes. Percentage values below the bars indicate the fraction of the values assigned to each bin for each region. The number of stations available for each region is reported in parentheses at the top of each panel.