# Peer review of "Low modeled ozone production suggests underestimation of precursor emissions (especially NOx) in Europe"

_Atmospheric Chemistry and Physics, 2017_

## Referee Comment (RC1) · Anonymous Referee #1 · 2 Oct 2017

General comments and overall quality

This paper is mainly focussed on the agreement between modelled and observed surface ozone in Europe in the year 2010 using various sensitivity model runs to identify reasons for mismatches between the modelled and observed levels of ozone. Other species like CO, SO2 and PM2.5 are also included but only to a very small extent. The work is presented in a clear and sound scientific way with no major errors, and overall this is as a robust and well-performed study with interesting findings that certainly deserves to be published. Some questions and comments are given in the following.

A few general comments: Neither the title, abstract or conclusions mention anything

about the additional species (SO2, CO and PM2.5) being included in the work. Furthermore, these species constitues a very small part of the paper and apparently with a fairly small implication for surface ozone which is the main focus of the paper. Thus, one could consider to take out these species completely. This is left to the authors (or the editor) to decide.

A main issue when doing comparisons between observed and modelled ozone is the question of how to treat vertical concentation gradients near the surface. During the summer season the effective dry deposition and uptake in vegetation will lead to signicifant gradients in ozone near the ground. Since the air intake of the ozone monitors mostly are at around 2 m the gas concentrations at that altitude in some way need to be related to the mean concentrations in the model's lowest layer, in this case around 20 m. Has this issue been considered and if not – how important could this effect be?

Underestimation of the high peak values is commonly seen in almost every model study. The authors should include some discussion on this general feature with references to a number of relevant modelling papers. Could it be that this artefact is reflecting the unavoidable smoothing (in emissions, meteorology etc) that all CTM relies on?

Specific comments

P3 L8 (and Fig 1). The definition of sub-regions and in particular sub-region 3 seems a bit odd. If the point is to divide Europe into areas with homogenous characteristics with respect to climate and air pollution statistics, then region 3 doesnt seem a very natural choice since it merges clean background sites (West coast of Ireland) with central European sites (e.g. Czech Republic). Apart from perhaps the most northern part, a latitudinally based definition of sub-regions is not very meaningful for Europe. Thus, it would make more sense to split region 3 into two regions or to create another set of sub-regions better reflecting climatological patterns. (See e.g. the PRUDENCE regions: http://ensemblesrt3.dmi.dk/quicklook/regions.html)

P3 L9. With the model top at 460 hPa, the domain seems shallow compared to the model setup that is normally used for regular modelling in Europe. The authors should inlude some sentences justifying this choice of vertical range.

P7 L4. For studying ozone peak values the time period 11-16 UT is selected "when the ozone production and mixing ratios often reach their maximum", the author states. This seems as a bit narrow and early to capture the highest ozone peak values. On average, for the entire 6-months summer season, 11-16 UT may be the peak period in some regions (see Fig 3). However, during high ozone episodes the peak values will often occur later in the day, and a period 12-18 UT would seem a more natural choice or even 14-20 UT.

P7 L14. Some details (geographical location and alitude) of the 8 stations with data on both T and O3 should be given, e.g. in a map.

P10 L17 (and Fig 4). How representative are the mean diurnal cycle of NO2 for this very large region? The header states that only 8 sites are included, and presumably (with some knowledge of the Airbase data) most of these sites are from the Northern UK?

Technical corrections

P2 L8-9. Consider rewriting this sentence: "Apart from the ozone precursor emissions, the other key driver of the surface ozone concentrations, as well as its chemistry, is the meteorology; from local to global scale". To state that meteorology is a "key driver" of surface ozone concentrations is somewhat meaningless without a few words explaing how met could affect ozone.

P2 L13-14. This sentence is imprecise. Although T is peaking in the afternoon, incoming solar radiation is not. Rewrite.

P2 L18. are -> is

P2 L19. This phrase should be reformulated and clearified: "The evaluation of modeled

ozone production from the ozone concentrations may not be a safe option."

P5 L4. This phrase should be reformulated: ". . . large discrepancies have been obvious . . ."

P7 L14: Rephrase this: ". . . surface stations which contain both temperature and ozone . . .". (The station doesn't "contain" temperature and ozone.

P8 L23. The text and the caption of Table 5 should explain for which time period (summer season?) these statistics were based on and for what type of data (hourly, afternoon means or something else?).

P10 L13. Typo: "overestimation" should be changed to "underestimation"

P10 L17. Rewrite. The word "now" doesn't seem appropriate. Change e.g. to "in this region" or something else.

Fig 5 and Fig 6. The time period (11-16 UT?) which the afternoon average is based on should be given in the Figure captions.

P12 L15. Rephrase this: ". . . seems to be the most effective scenario . . ." (The point is presumingly that the model scenario with increased emissions of both NOx and VOC is the scenario that gives the best fit with the observations).
* * *

---

## Referee Comment (RC2) · Anonymous Referee #1 · 9 Oct 2017

One additional remark:

Fig S10 in the Supplements shows the mean vertical O3 profile as measured by sondes and modelled for six European sites for summer 2010. All these profiles indicate a fairly marked underestimation of ozone by the model at the top model layer (about 4 km or a bit higher). Perhaps the authors could comment on this and on what possible consequenses this could have for the model performance at the surface. Does this reflect a systematic bias (underestimation) in the boundary condition of ozone at the upper boundary? This question should be seen together with my previous comment that the vertical extent of the model domain seems somewhat shallow compared to

standard CTMs that typically extend up to the tropopause. This question also leads to the question whether part of the systematic underestimation of ozone at the surface could be explained by the combination of too low concentration used as upper boundary conditions, a shallow vertical model range and uncertainties in the vertical exchange processes of ozone (the vertical gradients from the sondes differ markedly from the modelled ones).
* * *

---

## Referee Comment (RC3) · Anonymous Referee #2 · 29 Oct 2017

The authors' present an analysis assessing CAMx model under-predictions of ozone production in Europe, and arrive at the main finding that NOx emissions are likely under-predicted in existing emission inventories. Overall, the paper is well-written, clear, scientific methods appropriate, and in general findings/conclusions well-supported. I have a couple of critiques that hopefully will help strengthen this paper. First, I think the sensitivity analysis adjusting NOx emissions could be more specific to transportation emissions, rather than applied across total anthropogenic emissions. Second, the sensitivity analysis of wind speed seems to be in the opposite direction based on the model bias for this meteorological parameter. With revisions to the model test cases, I believe this manuscript could be considered for publication in Atmospheric

Chemistry & Physics.

General Comments

1. Section 2.3 ("Emissions"). The authors present a nice literature summary suggesting that transportation emissions of NOx are uncertain and may be underestimated by a factor of 2-4 (Page 5, Line 2). However, it appears that the authors' scaled all anthropogenic NOx emissions by a factor of 2 (Table 3) in sensitivity tests of the model. Based on Figure 2, scaling up road transportation emissions by a factor of 4 would roughly equal a factor of 2 increase in anthropogenic emissions. I suggest scaling the transportation sector only in the sensitivity analysis rather than all anthropogenic sources. First, it is not clear that point source emissions should exhibit uncertainties as large as the transportation sector. Second, the diurnal and day-of-week cycle in transportation emissions differ (Nassar et al., 2013) from point/area sources, which could affect diurnal and day-of-week patterns in the model and affect NO2 and O3 evaluations (Marr et al., 2002). Third, transportation emissions are likely more concentrated in urban cores relative to other sources of NOx (e.g., power generation/industry), which could affect the spatial distribution of emissions and model evaluations performed on rural background monitors (Page 6, Line 13).

2. Section 3.3 ("Sensitivity of ozone to meteorology"). The rationale behind increasing temperature by +4 degrees Celsius in the model (Table 3) makes sense based on systematic underestimates in temperature in the base case (Figure S3, also stated on Page 14, Line 3). However, why is wind speed reduced in the model rather than increased, when the model generally systematically underestimates wind speed in the base case (Table 4/Figure S4/Figure S6)? By increasing wind speed in the model, the ozone under-predictions will likely be worse, and a stronger argument can be made that meteorology is unlikely to explain the model discrepancies in relation to emissions.

Specific Comments

3. Section 2.3 (Page 4, Line 13). I think this paragraph could benefit from a description

of why anthropogenic VOC emissions are uncertain at a ∼50% level, similar to the proceeding discussion of why anthropogenic NOx emissions are uncertain. One sentence here seems too brief.

4. Section 3.1 (Page 10, Line 13). "...the overestimation of higher ones..." I believe the authors' mean *underestimation* here.

5. Section 3.2 (Page 12, Second Paragraph). A summary point at the end of the paragraph would be helpful here. It seems that the authors' might want to emphasize that NOx emissions need to be increased across most regions to improve results.

6. Section 3.2 (Page 12, Lines 21-22). It would help to label the slopes of the dashed grey lines in Figure 9, to help the reader more clearly discern the points made in this paragraph.

7. Section 3.3 ("Temperature"). Figures S3 and S5 seem inconsistent. While Figure S3 shows a general under-prediction of temperature by the base case model, Figure S5 seems to be showing a lot stations being over-predicted in the model (yellow and orange markers). I'm wondering if this related to the under-predicted sites being blocked out by the over-predicted sites in the coloring scheme. Suggest revising the presentation of Figure S5.

8. Section 3.3 (Page 14, Lines 25-31). These sentences do not seem to support the sensitivity test performed in the model where wind speeds are decreased, since: (i) the model seems to be doing well already (Line 25), (ii) most observations show a model under-prediction in wind speeds rather than over-prediction (Line 28), and (iii) the low wind speed conditions where the model over-predicts wind speeds comprise a minor fraction of observations (Line 30). Suggest revising this sensitivity test for wind speed, to increase rather than decrease in the model.

9. Section 4 (Last Paragraph). I think this last statement made here could be stronger by performing a sensitivity test of transportation NOx emissions only, which would be

in line with the literature suggesting that this sector is consistently underestimated in Europe (Annenberg et al., 2017; Karl et al., 2017). Rather than draw attention to all anthropogenic emission sources, it would be helpful to identify which sectors specifically need the most improvements in emission inventories.

References

Anenberg, S. C., et al. (2017). "Impacts and mitigation of excess diesel-related NOx emissions in 11 major vehicle markets." Nature 545(7655): 467-+.

Karl, T., et al. (2017). "Urban eddy covariance measurements reveal significant missing NOx emissions in Central Europe." Scientific Reports 7.

Marr, L. C. and R. A. Harley (2002). "Modeling the effect of weekday-weekend differences in motor vehicle emissions on photochemical air pollution in central California." Environmental Science & Technology 36(19): 4099-4106.

Nassar, R., et al. (2013). "Improving the temporal and spatial distribution of CO2 emissions from global fossil fuel emission data sets." Journal of Geophysical Research-Atmospheres 118(2): 917-933.

---

## Author Comment (AC1) · 13 Dec 2017

**Responses to the comments of anonymous referee #1**

We would like to thank for the comments which helped to improve our manuscript. Please find below your comments in blue, our responses in black and modifications in the revised manuscript related to technical or specific comments in italic and inside quotes. Additional text modifications have been done as most figures have been updated following your suggestions. All modifications are highlighted in the revised manuscript.

General comments and overall quality

This paper is mainly focussed on the agreement between modelled and observed surface ozone in Europe in the year 2010 using various sensitivity model runs to identify reasons for mismatches between the modelled and observed levels of ozone. Other species like CO, SO2 and PM2.5 are also included but only to a very small extent. The work is presented in a clear and sound scientific way with no major errors, and overall this is as a robust and well-performed study with interesting findings that certainly deserves to be published. Some questions and comments are given in the following.

A few general comments: Neither the title, abstract or conclusions mention anything about the additional species (SO2, CO and PM2.5) being included in the work. Furthermore, these species constitues a very small part of the paper and apparently with a fairly small implication for surface ozone which is the main focus of the paper. Thus, one could consider to take out these species completely. This is left to the authors (or the editor) to decide.

Thank you for this comment. We would like to keep them to show, even in a small extent, a more general model performance for a better comparison with past and future studies.

A main issue when doing comparisons between observed and modelled ozone is the question of how to treat vertical concentration gradients near the surface. During the summer season the effective dry deposition and uptake in vegetation will lead to signicifant gradients in ozone near the ground. Since the air intake of the ozone monitors mostly are at around 2 m the gas concentrations at that altitude in some way need to be related to the mean concentations in the model's lowest layer, in this case around 20 m. Has this issue been considered and if not – how important could this effect be?

The height of the first layer is indeed very important for the mixing and deposition processes. Menut et al. (2013) showed that setting the top of the model's lowest layer at a lower altitude (8 m compared to 40 m) results in consistent reductions of modeled (for August 2009 in Europe) ozone concentration by 3-12 μg m$^{-3}$ (~1.5-6 ppb) as ozone deposition is enhanced, especially over forested areas. Also, Travis et al. (2017) reported a decrease of ~3 ppb in ozone mixing ratios when accounting for the subgrid vertical gradient between their lowest model level (centered 60 m above ground) and their measurement altitude (10 m). This decrease in ozone was evident for their whole ozone concentration probability

distribution, shifting the distribution to the left (towards the lower ozone concentration values). In most of the applications, the regional models use a surface layer thickness between 20 and 90 m and in models with coarse surface layer (such as 90 m), usually a correction is implemented to represent surface concentrations (Bessagnet et al., 2016). In our application, the top of the lowest layer is at 20 m, but the ozone mixing ratios are calculated at the mid-point of each layer. So in this case the modeled ozone mixing ratios in the lowest layer are at 10 m which is reasonable for regional modeling. We added a sentence to clarify that:

*"We used 14 sigma layers going up to 460 hPa with the first layer being approximately 20 m thick. The concentrations are calculated at the mid-point of a given layer, so the modeled values of the first layer correspond to a height of approximately 10 m."*

Underestimation of the high peak values is commonly seen in almost every model study. The authors should include some discussion on this general feature with references to a number of relevant modelling papers. Could it be that this artefact is reflecting the unavoidable smoothing (in emissions, meteorology etc) that all CTM relies on?

Several studies (Valari and Menut, 2008; Markakis et al., 2015; Schaap et al., 2015; Kuik et al., 2016) have investigated the impact of the use of a finer model and emissions resolution on the CTM performance and they showed that it can improve the model performance for urban areas but has no significant impact on rural ones (for resolutions higher than the one used in this study (0.250 x 0.125)). Furthermore, it was shown that a finer resolution in the emissions and the model could lead to consistent overall reductions in ozone compared to the coarser resolution due to enhanced ozone titration by $NO_x$ in the urban areas. Therefore, we conclude that the model and emissions resolution of this study cannot be held responsible for the underestimation of high ozone mixing ratios.

We added a sentence to refer to other studies (Solazzo et al., 2012; Im et al., 2015) that report the model underestimation of high ozone concentrations for a variety of different models and parameterizations in Europe:

*"Similar model bias patterns as in this study were also reported by other studies for a variety of different models and parameterizations in Europe, the vast majority of which showed overestimation of low ozone concentrations and significant underestimation of the high ozone levels (Solazzo et al., 2012; Im et al., 2015)."*

Specific comments

P3 L8 (and Fig 1). The definition of sub-regions and in particular sub-region 3 seems a bit odd. If the point is to divide Europe into areas with homogenous characteristics with respect to climate and air pollution statistics, then region 3 doesnt seem a very natural choice since it merges clean background sites (West coast of Ireland) with central European sites (e.g. Czech Republic). Apart from perhaps the most northern part, a latitudinally based definition of sub-regions is not very meaningful for Europe. Thus, it would make more sense to split region 3 into two regions or to create another set of sub-

Thank you for this comment. We followed your suggestions and divided our domain in PRUDENCE regions, but with some modifications: i) the separation of the Benelux region for the reasons stated in the text, ii) we kept the Po Valley region instead of the more general "Alps" region in the PRUDENCE regions, iii) we grouped some stations in central France with the stations in central Europe (ME region), and iv) we grouped the few stations in northeastern Europe in the Eastern Europe region (EA) instead of Scandinavia region (SC). We updated all the respective figures and modified some of the text to properly describe and discuss the updated results.

P3 L9. With the model top at 460 hPa, the domain seems shallow compared to the model setup that is normally used for regular modelling in Europe. The authors should inlude some sentences justifying this choice of vertical range.

For the meteorological simulation with WRF we used 31 layers (up to 100 hPa) and for the air quality simulations with CAMx we used a selection of 14 out of 31 layers with higher resolution closer to the surface. We performed an additional simulation with the base case parameterization but this time including all 31 layers for June 2010 (and the last 2 weeks of May that were used as spin-up). Please note that this CAMx test simulation with the 31 layers was performed only for June and not for the whole summer (JJA). The results are shown in Fig. 1 below (we included this figure in the revised supplementary material as Fig. S1). The effect of extending our vertical range to higher altitude and increasing our vertical resolution leads to a small reduction in ozone (by ~1–2 ppb). This slightly improves the model performance in the lower ozone bins but also slightly worsens it for the higher ozone bins. Overall, the impact is quite small which is in line with other studies that have also reported a small sensitivity of the surface ozone to the refinement of the vertical mesh (Menut et al., 2013; Markakis et al., 2015). We added a sentence such that our choice of the vertical range seems more justified:

*"Additional tests showed that higher vertical resolution with layers up to 100 hPa would have a negligible effect on surface ozone (see Fig. S1) as also shown by other studies (Menut et al., 2013; Markakis et al., 2015)."*

[Figure]

**Figure. 1.** Effect of increasing vertical resolution (31 layers up to 100 hPa instead of 14 layers up to 460 hPa) on the mean afternoon (12:00–18:00 UTC) bias for surface $O_3$ mixing ratios in 8 European regions in June 2010. Percentage values below the bars indicate the fraction of the values assigned to each bin for each region. The number of stations available for each region is reported in parentheses at the top of each panel.

P7 L4. For studying ozone peak values the time period 11-16 UT is selected "when the ozone production and mixing ratios often reach their maximum", the author states. This seems as a bit narrow and early to capture the highest ozone peak values. On average, for the entire 6-months summer season, 11-16 UT may be the peak period in some regions (see Fig 3). However, during high ozone episodes the peak values will often occur later in the day, and a period 12-18 UT would seem a more natural choice or even 14-20 UT.

We agree and adapted the 12:00–18:00 UTC time interval, which corresponds to local summertime of 14:00–20:00 for the UTC+1 timezone countries where most of the stations (used in this study) are located.

P7 L14. Some details (geographical location and alitude) of the 8 stations with data on both T and O3 should be given, e.g. in a map.

We included a table in the supplement (Table S4) which provides the geographical location and elevation of these 8 stations.

P10 L17 (and Fig 4). How representative are the mean diurnal cycle of NO2 for this very large region? The header states that only 8 sites are included, and presumably (with some knowledge of the Airbase data) most of these sites are from the Northern UK?

After the implementation of the different regions separation, there are now only 3 stations in Scandinavia (SC region). Since this a small number of stations for such large region, the $NO_2$ results should not be interpreted as a robust representation of the whole SC region. We added a sentence to clarify that:

*"However, for BI and SC regions the $NO_2$ results should not be interpreted as a robust representation of the whole region due to the small number of sites (4 and 3 respectively) that are included."*

Technical corrections

P2 L8-9. Consider rewriting this sentence: "Apart from the ozone precursor emissions, the other key driver of the surface ozone concentrations, as well as its chemistry, is the meteorology; from local to global scale". To state that meteorology is a "key driver" of surface ozone concentrations is somewhat meaningless without a few words explaining how met could affect ozone.

We added a few words to explain how meteorology can affect ozone:

*"For example, on the local scale changes in shortwave solar radiation and temperature can directly influence the ozone photochemistry, and changes in wind speed or vertical mixing can lead to accumulation or dilution of the ozone precursor concentrations as well as ozone itself. On the global scale, changes in atmospheric circulation patterns can influence the continental transport of ozone concentrations and its precursors, the stratosphere–troposphere ozone exchange and the local meteorology."*

P2 L13-14. This sentence is imprecise. Although T is peaking in the afternoon, incoming solar radiation is not. Rewrite.

We corrected the sentence:

*"The peak values of surface ozone concentrations usually occur in the summer afternoon hours when the temperature reaches its diurnal maximum and the incoming solar radiation is still ample."*

P2 L18. are -> is

Corrected.

P2 L19. This phrase should be reformulated and clearified: "The evaluation of modeled ozone production from the ozone concentrations may not be a safe option."

The sentence was reformulated:

*"The evaluation of modeled ozone production by just comparing modeled ozone concentrations with measurements may be misleading, as an agreement between modeled and observed ozone concentrations might just be the result of compensating errors."*

P5 L4. This phrase should be reformulated: "… large discrepancies have been obvious …"

We corrected that phrase:

*"… large discrepancies have been observed …"*

P7 L14: Rephrase this: "… surface stations which contain both temperature and ozone …". (The station doesn't "contain" temperature and ozone.

We rephrased that part of the sentence:

*"… surface stations (see Table S4 for details), which have measurements of both temperature and ozone, …"*

P8 L23. The text and the caption of Table 5 should explain for which time period (summer season?) these statistics were based on and for what type of data (hourly, afternoon means or something else?).

We updated the caption of Table 5:

*"Table 5. Model performance evaluation for the daily mean concentrations of the chemical species in summer (JJA) 2010. The units for MB, MGE and RMSE are in ppb for the gas species and in $\mu g\ m^{-3}$ for the $PM_{2.5}$."*

We updated the text:

*"The overall model performance for the daily mean concentrations of the air pollutants in summer (JJA) 2010 (Table 5) was reasonably good."*

P10 L13. Typo: "overestimation" should be changed to "underestimation"

Corrected.

P10 L17. Rewrite. The word "now" doesn't seem appropriate. Change e.g. to "in this region" or something else.

We reformulated that part of the sentence:

*"…appears to be more pronounced…"*

Fig 5 and Fig 6. The time period (11-16 UT?) which the afternoon average is based on should be given in the Figure captions.

We updated all Figure captions to include the time interval that refers to the afternoon average (12:00–18:00 UTC).

P12 L15. Rephrase this: "… seems to be the most effective scenario …" (The point is presumingly that the model scenario with increased emissions of both NOx and VOC is the scenario that gives the best fit with the observations).

We changed that to:

*"… has the most effective improvement in the model performance for the BX region …"*

One additional remark:

Fig S10 in the Supplements shows the mean vertical O3 profile as measured by sondes and modelled for six European sites for summer 2010. All these profiles indicate a fairly marked underestimation of ozone by the model at the top model layer (about 4 km or a bit higher). Perhaps the authors could comment on this and on what possible consequenses this could have for the model performance at the surface. Does this reflect a systematic bias (underestimation) in the boundary condition of ozone at the upper boundary? This question should be seen together with my previous comment that the vertical extent of the model domain seems somewhat shallow compared to standard CTMs that typically extend up to the tropopause. This question also leads to the question whether part of the systematic underestimation of ozone at the surface could be explained by the combination of too low concentration used as upper boundary conditions, a shallow vertical model range and uncertainties in the vertical exchange processes of ozone (the vertical gradients from the sondes differ markedly from the modelled ones).

Our model top layer is at about 4.5-5 km a.g.l. Regarding the impact of a higher vertical range and resolution, we discussed it in detail in your specific comment (2) above where we showed that this impact was small. Regarding the ozone vertical profiles, the modeled vertical gradients seem weaker which might be an indication of increased vertical mixing. Similar issues were encountered and addressed by Travis et al. (2017), which were attributed to increased top-down mixing. When they applied some corrections to reduce the top-down mixing, the modeled and observed gradients were in much better agreement and the modeled surface ozone concentrations consistently decreased by 2–3 ppb as less ozone was transported from the upper layers to the surface. Therefore, this uncertainty could partially be responsible for the overestimation of the low ozone mixing ratios but a potential correction would also lead to a higher model underestimation in the high ozone bins. This issue of a potential increased top-down vertical mixing does not seem to be strongly related to our vertical structure choices, as the modeled ozone vertical gradients did not change significantly when we included all 31 layers as shown in Fig. 2 below. Regarding the large model underestimation of ozone in the upper layers, this could be related to a systematic bias in the boundary conditions, as has been also reported by other modeling studies (Giordano et al., 2015; Im et al., 2015; Solazzo et al., 2017). However, as we have shown in Fig. S12, the impact of increasing both the lateral and top boundary conditions

of ozone diminishes closer to the surface and into the interior of the domain. Furthermore, an increase in the ozone boundary conditions will increase the modeled ozone in all ozone bins and so the reduction of the model's underestimation in the high ozone bins will be accompanied by an overestimation in the rest of the ozone bins. In addition, with a reduced top-down mixing the effect of the higher altitude ozone concentrations on the surface will be smaller. Therefore, we conclude that the aforementioned uncertainties seem to be more related to the surface ozone overestimation than underestimation.

[Figure]

**Figure 2.** Effect of increasing vertical resolution (31 layers up to 100 hPa instead of 14 layers up to 460 hPa) on the ozone vertical profiles for 6 stations in June 2010. The number of ozonesondes available for each station is reported in parentheses at the top of each panel. Heights of 14 model layers are shown on both y-axes which are in logarithmic scale.

---

## Author Comment (AC2) · 13 Dec 2017

**Responses to the comments of anonymous referee #2**

We would like to thank for the comments that helped to improve our manuscript. Please find below your comments in blue, our responses in black and modifications in the revised manuscript related to technical or specific comments in italic and in quotes. Additional text modifications have been done as most figures have been updated following your suggestions. All modifications are highlighted in the revised manuscript.

The authors' present an analysis assessing CAMx model under-predictions of ozone production in Europe, and arrive at the main finding that NOx emissions are likely under-predicted in existing emission inventories. Overall, the paper is well written, clear, scientific methods appropriate, and in general findings/conclusions wellsupported. I have a couple of critiques that hopefully will help strengthen this paper. First, I think the sensitivity analysis adjusting NOx emissions could be more specific to transportation emissions, rather than applied across total anthropogenic emissions. Second, the sensitivity analysis of wind speed seems to be in the opposite direction based on the model bias for this meteorological parameter. With revisions to the model test cases, I believe this manuscript could be considered for publication in Atmospheric Chemistry & Physics.

General Comments

1. Section 2.3 ("Emissions"). The authors present a nice literature summary suggesting that transportation emissions of NOx are uncertain and may be underestimated by a factor of 2-4 (Page 5, Line 2). However, it appears that the authors' scaled all anthropogenic NOx emissions by a factor of 2 (Table 3) in sensitivity tests of the model. Based on Figure 2, scaling up road transportation emissions by a factor of 4 would roughly equal a factor of 2 increase in anthropogenic emissions. I suggest scaling the transportation sector only in the sensitivity analysis rather than all anthropogenic sources. First, it is not clear that point source emissions should exhibit uncertainties as large as the transportation sector. Second, the diurnal and day-of-week cycle in transportation emissions differ (Nassar et al., 2013) from point/area sources, which could affect diurnal and day-of-week patterns in the model and affect NO2 and O3 evaluations (Marr and Harley, 2002). Third, transportation emissions are likely more concentrated in urban cores relative to other sources of NOx (e.g., power generation/industry), which could affect the spatial distribution of emissions and model evaluations performed on rural background monitors (Page 6, Line 13).

Thank you for this suggestion. We included the suggested sensitivity test in our study. The sensitivity scenario is labeled "4traf_$NO_x$" where we scaled up only the road transport $NO_x$ emissions (SNAP 7) by a factor of 4. Indeed the behavior of the "4traf_$NO_x$" scenario is very similar to the "2$NO_x$" scenario, pointing out that probably the uncertainties in the $NO_x$ emissions that are related to the transportation sector are more responsible for the model bias patterns examined in this study. We updated all the respective figures and modified the text to properly describe and discuss the updated results.

2. Section 3.3 ("Sensitivity of ozone to meteorology"). The rationale behind increasing temperature by +4 degrees Celsius in the model (Table 3) makes sense based on systematic underestimates in temperature in the base case (Figure S3, also stated on Page 14, Line 3). However, why is wind speed reduced in the model rather than increased, when the model generally systematically underestimates wind speed in the base case (Table 4/Figure S4/Figure S6)? By increasing wind speed in the model, the ozone under-predictions will likely be worse, and a stronger argument can be made that meteorology is unlikely to explain the model discrepancies in relation to emissions.

We agreed and followed your suggestion by applying a sensitivity test with double the wind speed ("WSx2"), but we also kept the previous sensitivity test of "WS/2" to investigate also the overestimation of the low wind speed ($< 2$ m s$^{-1}$) in the regions where the fraction of the low wind speed overestimation compared to the total bias is not very small. We updated all the respective figures and we edited or added some text to properly describe and discuss the updated results.

Specific Comments

3. Section 2.3 (Page 4, Line 13). I think this paragraph could benefit from a description of why anthropogenic VOC emissions are uncertain at a ~50% level, similar to the proceeding discussion of why anthropogenic NOx emissions are uncertain. One sentence here seems too brief.

We discussed some possible sources of uncertainty for the anthropogenic VOC emissions:

*"The VOC emission uncertainties can be due to a number of reasons such as: i) the small number of measured vehicles for the transportation sector, since the VOC species resolution rely on measurements, ii) not enough available measurement data for the combustion-, process-, and production-related emissions compared to the much higher number of individual emission sources, iii) the large variety of the VOC compositions in the used solvents, iv) the measurement uncertainties (Theloke and Friedrich, 2007) "*

4. Section 3.1 (Page 10, Line 13). "… the overestimation of higher ones …" I believe the authors' mean *underestimation* here.

Corrected.

5. Section 3.2 (Page 12, Second Paragraph). A summary point at the end of the paragraph would be helpful here. It seems that the authors' might want to emphasize that NOx emissions need to be increased across most regions to improve results.

 A sentence was added as a summary point at the end of that paragraph:

*"Overall, our emission-sensitivity analysis indicates that the $NO_x$ emissions, especially from the transportation sector (SNAP 7) in central, eastern and southern Europe might be too low in the emission inventories."*

6. Section 3.2 (Page 12, Lines 21-22). It would help to label the slopes of the dashed grey lines in Figure 9, to help the reader more clearly discern the points made in this paragraph.

We modified the regions in the European domain and it is now even more clear that different regions have different model bias patterns and/or different responses to the sensitivity tests. Therefore, we removed the dashed grey line form Fig. 9, as we believe that this line will represent more the larger regions (ME and BX) and for most of the emissions sensitivity tests these regions have different responses, which can compensate each other. We think that it is better for the reader to extract more consistent information from this figure by examining the position of the region-colored scatter points around the 1:1, 1:2 and 2:1 lines.

7. Section 3.3 ("Temperature"). Figures S3 and S5 seem inconsistent. While Figure S3 shows a general under-prediction of temperature by the base case model, Figure S5 seems to be showing a lot stations being over-predicted in the model (yellow and orange markers). I'm wondering if this related to the under-predicted sites being blocked out by the over-predicted sites in the coloring scheme. Suggest revising the presentation of Figure S5.

We modified the Figures S4 and S6 (i.e. S3 and S5 in the original manuscript) changing the coloring scheme and shrinking the markers in Fig. S6. With the new region separation the comparison of the temperature bias in Eastern Europe (EA region) between the Figures S4 and S6 is now more clear. The temperature overestimation that is observed in Fig. S6 is also depicted in Fig. S4. We believe that Figures S4 and S6 look now consistent.

8. Section 3.3 (Page 14, Lines 25-31). These sentences do not seem to support the sensitivity test performed in the model where wind speeds are decreased, since: (i) the model seems to be doing well already (Line 25), (ii) most observations show a model under-prediction in wind speeds rather than over-prediction (Line 28), and (iii) the low wind speed conditions where the model over-predicts wind speeds comprise a minor fraction of observations (Line 30). Suggest revising this sensitivity test for wind speed, to increase rather than decrease in the model.

We responded to your general comment (2) about doubling the wind speed. Also, we replaced the wind speed normalized mean bias and gross error with the not normalized ones in Fig. S5, as we believe that the non-relative statistical metric for the bias and gross error gives a more robust overview of the model performance. In addition, the model evaluation in Fig. S5 is performed against observational data bins and so a general overview of the model performance in relative terms can also be seen.

9. Section 4 (Last Paragraph). I think this last statement made here could be stronger by performing a sensitivity test of transportation NOx emissions only, which would be in line with the literature suggesting that this sector is consistently underestimated in Europe (Anenberg et al., 2017; Karl et al., 2017). Rather than draw attention to all anthropogenic emission sources, it would be helpful to identify which sectors specifically need the most improvements in emission inventories.

Thank you for your suggestion. As we wrote above, we followed your suggestion and performed the sensitivity test with traffic $NO_x$ emissions.

**References**

Theloke, J., and Friedrich, R.: Compilation of a database on the composition of anthropogenic VOC emissions for atmospheric modeling in Europe, Atmos. Environ., 41, 4148-4160, doi:10.1016/j.atmosenv.2006.12.026, 2007.

---

## Author Response (AR2)

**Response to the suggested minor revisions by the Co-editor**

We would like to thank the Co-editor for his comments and suggestions which helped to improve our manuscript. Please find below our modifications in the revised manuscript related to your comments in italic and inside quotes. All modifications are highlighted in the revised manuscript.

We re-formulated the abstract, page 1, lines 25-28, as follows:

*"Although increasing only the traffic $NO_x$ emissions by a factor of 4 gave very similar results as the doubling of all $NO_x$ emissions, the first scenario is more consistent with the uncertainties reported by other studies than the latter suggesting that high uncertainties in $NO_x$ emissions might originate mainly from the road-transport sector rather than other sectors."*

We re-formulated the conclusions, page 18, lines 27-32 and page 19, line 1, as follows:

*"Increasing only traffic $NO_x$ emissions by a factor of 4 had almost the same impact as doubling all $NO_x$ emissions. However, as discussed in Sect. 2.3, previous investigations indicate higher uncertainties in $NO_x$ emissions from the road-transport compared to other sectors. Therefore, the 4traf_$NO_x$ scenario is more consistent with the previous studies than the 2$NO_x$ scenario, suggesting that high uncertainties in the $NO_x$ emissions from road-transport are more likely to be the main reason for underestimated ozone production rather than uncertainties in emissions from other sectors. For the less polluted British Isles (BI) and Scandinavia (SC) regions no emission adjustment was necessary."*